# ADVERSARIAL ATTACKS ON COMBINATORIAL MULTI-ARMED BANDITS

## ABSTRACT

We study reward poisoning attacks on Combinatorial Multi-armed Bandits (CMAB). We first provide a sufficient and necessary condition for the attackability of CMAB, which depends on the intrinsic properties of the corresponding CMAB instance such as the reward distributions of super arms and outcome distributions of base arms. Additionally, we devise an attack algorithm for attackable CMAB instances. Contrary to prior understanding of multi-armed bandits, our work reveals a surprising fact that the attackability of a specific CMAB instance also depends on whether the bandit instance is known or unknown to the adversary. This finding indicates that adversarial attacks on CMAB are difficult in practice and a general attack strategy for any CMAB instance does not exist since the environment is mostly unknown to the adversary. We validate our theoretical findings via extensive experiments on real-world CMAB applications including probabilistic maximum covering problem, online minimum spanning tree, cascading bandits for online ranking, and online shortest path.

## 1 INTRODUCTION

Multi-armed bandits (MAB) (Auer, 2002) is a classic framework of sequential decision-making problems that has been extensively studied (Lattimore & Szepesvári, 2020; Slivkins et al., 2019). In each round, the learning agent selects one out of $m$ arms and observes its reward feedback which follows an unknown reward distribution. The goal is to maximize the cumulative reward, which requires the agent to balance exploitation (selecting the arm with the highest average reward) and exploration (exploring arms that have high potential but have not been played enough).

Combinatorial multi-armed bandits (CMAB) is a generalized setting of original MAB with many real-world applications such as online advertising, ranking, and influence maximization (Liu & Zhao, 2012; Kveton et al., 2015; Chen et al., 2016; Wang & Chen, 2017). In CMAB, the agent chooses a combinatorial action (called a super arm) over the $m$ base arms in each round, and observes outcomes of base arms triggered by the action as feedback, known as the *semi-bandit* feedback. The exploration-exploitation trade-off in CMAB is extremely hard compared to MAB because the number of candidate super arms could be *exponential* in $m$.

Recent studies showed that MAB and its variants are vulnerable to adversarial attacks, especially poisoning attacks (Jun et al., 2018; Liu & Shroff, 2019; Wang et al., 2022; Garcelon et al., 2020). Under such attacks, an adversary observes the pulled arm and its reward feedback, and then modifies the reward to misguide the bandit algorithm to pull a target arm that is in the adversary's interest. Specifically, the adversary aims to spend attack cost sublinear in time horizon $T$ to modify rewards, i.e., $o(T)$ cost, such that the bandit algorithm pulls the target arm *almost all the time*, i.e., $T - o(T)$ times. Liu & Shroff (2019) showed that no-regret MAB algorithms can be efficiently attacked under any problem instance, while Wang et al. (2022) showed that there exist instances of linear bandits that cannot be attacked without linear cost, indicating such instances are *intrinsically robust* to adversarial attacks even provided vanilla bandit algorithms.

Due to the wide applicability of CMAB, understanding its vulnerability and robustness to poisoning attacks is increasingly important. Applying the MAB concept of *attackability* to CMAB is tempting (Liu & Shroff, 2019; Wang et al., 2022), but it leads to a sublinear cost bound in $T$ but exponential in the number of base arms $m$. This approach follows the same attack strategy as in MAB. However, in practice, the exponential cost in $m$ can exceed $T$, resulting in vacuous results. Therefore, the

original *attackability* notion is insufficient, and we have the following research question(RQ): *What is a good notion to capture the vulnerability and robustness of CMAB?*

## 1.1 OUR CONTRIBUTION

To answer the research question, we propose the definition of *polynomial attackability* (Section 3): when the attack is "successful", the cost upper bound should not only be sublinear in time horizon $T$, but also polynomial in the number of base arms $m$ and other factors. Following the definition of polynomial attackability, we propose the first study on adversarial attacks on combinatorial multi-armed bandits under the reward poisoning attack (Jun et al., 2018; Liu & Shroff, 2019; Wang et al., 2022). We provide a sufficient and necessary condition for polynomial attackability of CMAB, which depends on the bandit environment, such as the reward distributions of super arms and outcome distributions of base arms (Section 3). We analyze the attackability of several real-world CMAB applications including cascading bandits for online learning to rank (Kveton et al., 2015), online shortest path (Liu & Zhao, 2012), online minimum spanning tree (Kveton et al., 2014), and probabilistic maximum coverage problem (Chen et al., 2016) (a special instance of online influence maximization (Wen et al., 2017)). Our results can also be applied to simple reinforcement learning settings (Section 3), and to the best of our knowledge, get the first "instance"-level attackability result on reinforcement learning.

Perhaps surprisingly, we discovered that for the same CMAB instance, polynomial attackability is not always the same, but is conditioned on whether the bandit environment is known or unknown to the adversary (Section 4). We constructed a *hard example* such that the instance is polynomially attackable if the environment is known to the adversary but polynomially unattackable if it is unknown to the adversary in advance. This hardness result suggests that adversarial attacks on CMAB may be extremely difficult in practice and a general attack strategy for any CMAB instance does not exist since the environment is mostly unknown to the adversary.

Finally, numerical experiments are conducted to verify our theory (Section 5).

## 1.2 RELATED WORKS

**Adversarial attacks on bandits and reinforcement learning** Reward poisoning attacks on bandit algorithms were first studied in the stochastic multi-armed bandit setting (Jun et al., 2018; Liu & Shroff, 2019), where an adversary can always force the bandit algorithm to pull a target arm linear times only using a logarithmic cost. Garcelon et al. (2020) studied attacks on linear contextual bandits. The notion of attackability is first termed by Wang et al. (2022), where they studied the attackability of linear stochastic bandits. Adversarial attacks on reinforcement learning have been studied under white box (Rakhsha et al., 2020; Zhang et al., 2020) and black-box setting (Rakhsha et al., 2021; Rangi et al., 2022). Specifically, Rangi et al. (2022) showed there exist unattackable episodic RL instances with reward poisoning attacks. However, none of the existing work analyzed the attackability of a given instance. Besides reward poisoning attacks, other threat models such as environment poisoning attacks Rakhsha et al. (2020); Sun et al. (2021); Xu et al. (2021); Rangi et al. (2022) and action poisoning attacks Liu & Lai (2020) were also being studied. We focus on reward poisoning attacks in this paper and leave investigation on other threat models as future work.

**Corruption-tolerant bandits** Another line of work studies the robustness of bandit algorithms against poisoning attacks, also known as corruption-tolerant bandits. Lykouris et al. (2018); Gupta et al. (2019) proposed robust MAB algorithms under an oblivious adversary who determines the manipulation before the bandit algorithm pulls an arm. Dong et al. (2022) proposed a robust CMAB algorithm under strategic manipulations where the corruptions are limited to only increase the outcome of base arms in semi-bandit feedback. This is weaker than our threat model as we allow the adversary to increase or decrease the outcome of base arms.

## 2 PRELIMINARY

### 2.1 COMBINATORIAL SEMI-BANDIT

In this section, we introduce our model for the combinatorial semi-bandit (CMAB) problem. The CMAB model is mainly based on Wang & Chen (2017), which handles nonlinear reward functions,

approximate offline oracle, and probabilistically triggered base arms. A CMAB problem (Wang & Chen, 2017) can be considered a game between a *player* and an *environment*. We summarize the important concepts involved in a CMAB problem here:

- **Base arm:** The environment has $m$ base arms $[m] = 1, 2, \ldots, m$ associated with random variables following a joint distribution $\mathcal{D}$ over $[0,1]^m$. At each time $t$, random outcomes $\boldsymbol{X}^{(t)} = (X_1^{(t)}, X_2^{(t)}, \ldots, X_m^{(t)})$ are sampled from $\mathcal{D}$. The unknown mean vector $\boldsymbol{\mu} = (\mu_1, \mu_2, \ldots, \mu_m)$ where $\mu_i = \mathbb{E}_{X \sim \mathcal{D}}[X_i^{(t)}]$ represents the means of the $m$ base arms.

- **Super arm:** At time $t$, the player selects a base arm set $\mathcal{S}^{(t)}$ from action space $\mathbb{S}$ (which could be infinite) based on the feedback from the previous rounds. The base arm set, referred to as a ***super arm***, is composed of individual ***base arms***.

- **Probabilistically triggered base arms:** When the player selects a super arm $\mathcal{S}^{(t)}$, a random subset $\tau_t \subseteq [m]$ of base arms is triggered, and the outcomes $X_i^{(t)}$ for $i \in \tau_t$ are observed as feedback. $\tau_t$ is sampled from the distribution $\mathcal{D}^{\text{trig}}(\mathcal{S}^{(t)}, \boldsymbol{X}^{(t)})$, where $\mathcal{D}^{\text{trig}}(\mathcal{S}, \boldsymbol{X})$ is the probabilistic triggering function on the subsets $2^{[m]}$ given $\mathcal{S}$ and $\boldsymbol{X}$. We denote the probability of triggering arm $i$ with action $\mathcal{S}$ as $p_i^{\mathcal{D}, \mathcal{S}}$ where $\mathcal{D}$ is the environment triggering distribution. The set of base arms that can be triggered by $\mathcal{S}$ under $\mathcal{D}$ is denoted as $\mathcal{O}_{\mathcal{S}} = \{i \in [m] : p_i^{\mathcal{D}, \mathcal{S}} > 0\}$.

- **Reward:** The player receives a nonnegative reward $R(\mathcal{S}^{(t)}, \boldsymbol{X}^{(t)}, \tau_t)$ determined by $\mathcal{S}^{(t)}, \boldsymbol{X}^{(t)}$, and $\tau_t$. The objective is to select an arm $\mathcal{S}^{(t)}$ in each round $t$ to maximize cumulative reward. We assume that $\mathbb{E}[R(\mathcal{S}^{(t)}, \boldsymbol{X}^{(t)}, \tau_t)]$ depends on $\mathcal{S}^{(t)}, \boldsymbol{\mu}_t$, and denote $r_{\mathcal{S}}(\boldsymbol{\mu}) := \mathbb{E}_{\boldsymbol{X}}[R(\mathcal{S}, \boldsymbol{X}, \tau)]$ as the expected reward of super arm $\mathcal{S}$ with mean vector $\boldsymbol{\mu}$. This assumption is similar to Chen et al. (2016); Wang & Chen (2017), and holds when variables $X_i^{(t)}$ are independent Bernoulli random variables. We define $\text{opt}\boldsymbol{\mu} := \sup_{\mathcal{S} \in \mathbb{S}} r_{\mathcal{S}}(\boldsymbol{\mu})$ as the maximum reward given $\boldsymbol{\mu}$.

In summary, a *CMAB problem instance* with probabilistically triggered arms can be described as a tuple $([m], \mathbb{S}, \mathcal{D}, \mathcal{D}^{\text{trig}}, R)$. We introduce the following assumptions on the reward function, which are standard assumptions commonly used in the CMAB problem.

**Assumption 2.1** (Monotonicity). *For any $\boldsymbol{\mu}$ and $\boldsymbol{\mu}'$ with $\boldsymbol{\mu} \preceq \boldsymbol{\mu}'$ (dimension-wise), for any super arm $\mathcal{S} \in \mathbb{S}$, $r_{\mathcal{S}}(\boldsymbol{\mu}) \leq r_{\mathcal{S}}(\boldsymbol{\mu}')$.*

**Assumption 2.2** (1-Norm TPM Bounded Smoothness). *For any two distributions $\mathcal{D}, \mathcal{D}'$ with expectation vectors $\boldsymbol{\mu}$ and $\boldsymbol{\mu}'$ and any super arm $\mathcal{S} \in \mathbb{S}$, there exists a $B \in \mathbb{R}^+$ such that,*

$$|r_{\mathcal{S}}(\boldsymbol{\mu}) - r_{\mathcal{S}}(\boldsymbol{\mu}')| \leq B \sum_{i \in [m]} p_i^{\mathcal{D}, \mathcal{S}} |\mu_i - \mu_i'|.$$

**CUCB algorithm** The combinatorial upper confidence bound (CUCB) algorithms (Chen et al., 2013; Wang & Chen, 2017) are a series of algorithms devised for the CMAB problem. Key ingredients of a typical CUCB algorithm include (1) optimistic estimations of the expected value of the base arms $\boldsymbol{\mu}$ and (2) a computational oracle that takes the expected value vector $\boldsymbol{\mu}$ as input and returns an optimal or close-to-optimal super arm. For example, an $(\alpha, \beta)$-approximation oracle can ensure that $Pr(r_{\mathcal{S}}(\boldsymbol{\mu}) \geq \alpha \cdot \text{opt}_{\boldsymbol{\mu}}) \geq \beta$. A typical CUCB algorithm proceeds in the following manner: in each round, the player tries to construct tight upper confidence bound (UCB) on the expected outcome based on historical observations, and feeds the UCBs to the computational oracle; the oracle yields which super arm to select. When combined with the two assumptions above, a legitimate CUCB algorithm's regret, or the $(\alpha, \beta)$-approximation regret, can typically be upper bounded.

## 2.2 THREAT MODEL

We consider reward poisoning attack as the threat model (Jun et al., 2018; Liu & Shroff, 2019; Garcelon et al., 2020; Wang et al., 2022). Under such threat model, a malicious adversary has a set $\mathcal{M}$ of target super arms in mind and the goal of the adversary is to misguide the player into pulling any of the target super arm linear times in the time horizon $T$. At each round $t$, the adversary observes the pulled super arm $\mathcal{S}^{(t)}$, the outcome of the base arms $\boldsymbol{X}^{(t)} = \{X_i^{(t)}\}_{i \in \tau_t}$, and its reward feedback $R(\mathcal{S}^{(t)}, \boldsymbol{X}^{(t)}, \tau_t)$. The adversary modifies the outcome of base arm from $X_i^{(t)}$ to $\tilde{X}_i^{(t)}$ for all $i \in \tau_t$. $\tilde{\boldsymbol{X}}^{(t)} := \{\tilde{X}_i^{(t)}\}_{i \in \tau_t}$ is thus the 'corrupted return' of the observed base arms. The cost of the attack is defined as $C(T) = \sum_{t=1}^{T} \|\tilde{\boldsymbol{X}}^{(t)} - \boldsymbol{X}^{(t)}\|_0$.

## 2.3 Selected applications of CMAB

**Online minimum spanning tree**   Consider an undirected graph $\mathcal{G} = (V, E)$. Every edge $e = (u, v)$ in the graph has a cost realization $X_{u,v}^{(t)} \in [0, 1]$ drawn from a distribution with mean $\mu_{u,v}$ at each time slot $t$. Online minimum spanning tree problem is to select a spanning tree $\mathcal{S}^{(t)}$ at every time step $t$ on an unknown graph in order to minimize the cumulative (pseudo) regret defined as $\mathcal{R} = \sum_{t=1}^{T} \mathbb{E} \big[ \sum_{(u,v) \in \mathcal{S}^{(t)}} X_{u,v}^t - \sum_{(u,v) \in \mathcal{S}^*} X_{u,v}^t \big]$, where $\mathcal{S}^*$ is the minimum spanning tree given the edge cost $\mu_e$ for every edge $e$.

**Online shortest path**   Consider an undirected graph $\mathcal{G} = (V, E)$. Every edge $e = (u, v)$ in the graph has a cost realization $X_{u,v}^{(t)} \in [0, 1]$ drawn from a distribution with mean $\mu_{u,v}$ at each time slot $t$. Given the parameters $\mu_e$, the shortest path problem with a source $\mathbf{s}$ and a destination $\mathbf{t}$ is to select a path from $\mathbf{s}$ to $\mathbf{t}$ with minimum cost, i.e., to solve the following problem $\min_{\mathcal{S}} \sum_{(u,v) \in \mathcal{S}} \mu_{u,v}$ where $\mathcal{S}$ is a path from $\mathbf{s}$ to $\mathbf{t}$. The online version is to select a path $\mathcal{S}^{(t)}$ from $\mathbf{s}$ to $\mathbf{t}$ at time $t$ on graph $\mathcal{G} = (V, E)$ in order to minimize the cumulative regret.

**Cascading bandit**   Cascading bandit is a popular model for online learning to rank with Bernoulli click feedback under cascade click model (Kveton et al., 2015; Vial et al., 2022). There is a set of items (base arms) $[m] = \{1, 2, 3, \cdots, m\}$. At round $t$, the bandit algorithm recommends a list of $K < m$ items $[a_{t,1}, a_{t,2}, \cdots a_{t,K}]$ as a super arm to the user from which the user clicks on the first attractive item (if any), and stops examining items after clicking. The user selects item $j$ from the list with probability $\mu_j$, which should be estimated online by the algorithm. The cascading bandit problem can be reduced to CMAB with probabilistic triggered arms (Wang & Chen, 2017).

**Probabilistic maximum coverage**   Given a weighted bipartite graph $\mathcal{G} = (L, R, E)$ where each edge $(u, v)$ has a probability $\mu_{u,v}$, the probabilistic maximum coverage (PMC) is to find a set $S \subseteq L$ of size $k$ that maximizes the expected number of activated nodes in $R$, where a node $v \in R$ can be activated by a node $u \in S$ with an independent probability of $\mu_{u,v}$. In the online version of PMC, $\mu_{u,v}$ are unknown. The algorithm estimates $\mu_{u,v}$ online and minimizes the pseudo-regret.

## 3 Polynomial Attackability of CMAB Instances

Previous literature defined the following *attackability* notion for MAB instances (Jun et al., 2018; Liu & Shroff, 2019; Garcelon et al., 2020; Wang et al., 2022): for any no-regret algorithm $\mathcal{A}$ (the regret of $\mathcal{A}$ is $o(T)$ for large enough $T$), the attack can only use sublinear cost $C(T) = o(T)$ and fool the algorithm $\mathcal{A}$ to play the arms in target set $\mathcal{M}$ for $T - o(T)$ times. It is worth noting that Wang et al. (2022) is the first to observe that certain linear MAB instances are *unattackable* without linear cost, suggesting *intrinsic robustness* of such linear bandit instances. However the notion of *attackability* needs to be modified in the CMAB framework, since the number of super arms is exponential and will lead to impractical guarantees. In this work, we propose to consider the *polynomial attackability* of CMAB.

For a particular CMAB instance $([m], \mathbb{S}, \mathcal{D}, \mathcal{D}^{\text{trig}}, R)$, we define $p^* := \inf_{\mathcal{S} \in \mathbb{S}, i \in \mathcal{O}_{\mathcal{S}}} \{p_i^{\mathcal{D}, \mathcal{S}}\}$.

**Definition 3.1** (Polynomially attackable[1]).   A CMAB instance is polynomially attackable with respect to a set of super arms $\mathcal{M}$, if for any learning algorithm with regret $O(\text{poly}(m, 1/p^*, K) \cdot T^{1-\gamma})$ with high probability for some constant $\gamma > 0$, there exists an attack method with constant $\gamma' > 0$ that uses at most $T^{1-\gamma'}$ attack cost and misguides the algorithm to pull super arm $\mathcal{S} \in \mathcal{M}$ for $T - T^{1-\gamma'}$ times with high probability for any $T \geq T^*$, where $T^*$ polynomially depends on $m, 1/p^*, K$.

**Definition 3.2** (Polynomially unattackable).   A CMAB instance is polynomially unattackable with respect to a set of super arms $\mathcal{M}$ if there exists a learning algorithm $\mathcal{A}$ with regret $O(\text{poly}(m, 1/p^*, K) \cdot T^{1-\gamma})$ with high probability for some constant $\gamma > 0$, such that for any attack method with constant $\gamma' > 0$ that uses at most $T^{1-\gamma'}$ attack cost, the algorithm $\mathcal{A}$ will pull super arms $\mathcal{S} \in \mathcal{M}$ for at most $T/2$ times with high probability for any $T \geq T^*$, where $T^*$ polynomially depends on $m, 1/p^*, K$.

---

[1] We use the terms *attackable* and *polynomially attackable* interchangeably in the following discussion.

*Remark* 3.3 (Conventional attackability definition vs. polynomially attackable vs. polynomially unattackable). 1. Compared to conventional attackability definitions for $k$-armed stochastic bandit instance, both Definition 3.1 and Definition 3.2 require a polynomial dependency on $m$. 2. Note that the 'polynomially unattackable' notion is stronger than 'not polynomially attackable'.

*Remark* 3.4 (Polynomial dependency). $T^*$'s dependency on $m, 1/p^*, K$ is important to CMAB. Otherwise, the problem reduces to a vanilla MAB with an exponentially large number of arms.

**Definition 3.5** (Gap). For each super arm $\mathcal{S}$, we define the following gap

$$\Delta_{\mathcal{S}} := r_{\mathcal{S}}(\boldsymbol{\mu}) - \max_{\mathcal{S}' \neq \mathcal{S}} r_{\mathcal{S}'}(\boldsymbol{\mu} \odot \mathcal{O}_{\mathcal{S}}).$$

where $\odot$ is the element-wise product. For a set $\mathcal{M}$ of super arms, we define the gap of $\mathcal{M}$ as $\Delta_{\mathcal{M}} = \max_{\mathcal{S} \in \mathcal{M}} \Delta_{\mathcal{S}}$.

---

**Algorithm 1** Attack algorithm for CMAB instance

---

**Require:** Target arm set $\mathcal{M}$ such that $\Delta_{\mathcal{M}} > 0$, CMAB algorithm Alg.
 1: Find a super arm $\mathcal{S} \in \mathcal{M}$ such that $\Delta_{\mathcal{S}} > 0$.
 2: **for** $t = 1, 2, \ldots, T$ **do**
 3:     Alg returns super arm $\mathcal{S}^{(t)}$.
 4:     Adversary returns $\tilde{X}_i^t = 0$ for all $i \in \tau^{(t)} \setminus \mathcal{O}_{\mathcal{S}}$ and keep other outcome $X_i^t$ unchanged.
 5: **end for**

---

**Theorem 3.6** (Polynomial attackability of CMAB). *Given a particular CMAB instance and the target set of super arms $\mathcal{M}$ to attack. If $\Delta_{\mathcal{M}} > 0$, then the CMAB instance is polynomially attackable. If $\Delta_{\mathcal{M}} < 0$, the instance is polynomially unattackalble.*

We do not consider the special case of $\Delta_{\mathcal{M}} = 0$ in Theorem 3.6 as the condition is ill-defined for attackability: it relies on how the CMAB algorithm breaks the tie and we omit the case in the theorem.

Theorem 3.6 provides a *sufficient and necessary* condition for the CMAB instance to be polynomial attackable. Specifically, we prove the sufficiency condition ($\Delta_{\mathcal{M}} > 0$) by constructing Algorithm 1 as an attack that spends $\text{poly}(m, 1/p^*, K) \cdot m \cdot (1/\Delta_{\mathcal{S}^*}) \cdot T^{1-\gamma}$ attack cost to pull target super arms linear time. The main idea is to reduce the reward of base arms that are not associated with the target super arms to 0. On the other hand, we show that $\Delta_{\mathcal{M}} < 0$ leads to polynomial unattackable instance, which is stronger than and covers *not polynomial attackable*, and serves as the necessary condition.

Note that in Theorem 3.6, we do not specify the attack cost since the cost depends on not only the CMAB instance but also the victim CMAB algorithm. If we specify the victim algorithm to be CUCB, which is the most general algorithm for solving CMAB problem, we can have the following results directly from the proof of Theorem 3.6.

**Corollary 3.7.** *Given a particular CMAB instance and the target set of super arms $\mathcal{M}$ to attack. If $\Delta_{\mathcal{M}} > 0$ and the victim algorithm is chosen to be CUCB, then the attack cost can be bounded by $\text{poly}(m, K, 1/p, 1/\Delta) \cdot \log(T)$.*

In general, the probability $p^*$ may be exponentially small, and there already exists analysis for the combinatorial semi-bandit (Wang & Chen, 2017) that can remove this $p^*$ dependence. However, there are still some differences between the original CMAB setting and our attack setting. The following example shows the necessity of the term $p^*$: if we remove the dependency on $1/p^*$ in Definition 3.1 and 3.2, we show that there exist instances such that it is neither "polynomial attackable" nor "polynomial unattackable". Consider the case that there are 3 base arms $a, b, c$. $\mu_a = \mu_b = 1/2, \mu_c = 1/4$, and two super arms $\mathcal{S}_1, \mathcal{S}_2$. Assume that when you pull $\mathcal{S}_1$, you observe base arm $b$ with probability 1, and $a$ with probability $p < \frac{1}{2}$, and if you pull $\mathcal{S}_2$, you observe $a, c$ with probability 1. Thus, the reward of $\mathcal{S}_1$ is $R_b + \mathbb{I}[\mathcal{E}_a]R_a$ where $\mathcal{E}_a$ is the event where $a$ is triggered, and the reward of $\mathcal{S}_2$ is $R_a + R_c$. Now note that if we set the time scale $T$ large enough, we know that CUCB will not play $\mathcal{S}_1$ for $T - o(T)$ time. However if $T$ is not that large, it is still possible to misguide CUCB to play $\mathcal{S}_1$ for $T - o(T)$ times. The attack strategy is that: whenever $a$ is observed, we first set the reward for $a$ to be 0 and let UCB $(a) < 1/4$. Then CUCB will pull $\mathcal{S}_1$. However if $\mathcal{S}_1$ is not pulled by $1/p$ times, it may not observe base arm $a$ and thus $\mathcal{S}_1$ can still be played for $T - o(T)$ times for $T$ not large enough (say, not dependent on $1/p$).

Following the Theorem 3.6, we now analyze the attackability of several practical CMAB problems.

**Corollary 3.8.** *In online minimum spanning tree problem, for any super arm $\mathcal{S}$, $\Delta_{\mathcal{S}} \geq 0$. In cascading bandit problem, for any super arm $\mathcal{S}$, $\Delta_{\mathcal{M}=permutation(\mathcal{S})} \geq 0$.*

Notice that if the goal of the CMAB problem is to minimize the objective (cost) such as online minimum spanning tree, we change line 3 of Algorithm 1 to $\bar{X}_i^T = 1$ where we modify the base arm into having highest cost.

Corollary 3.8 is relatively straightforward. First, we know that every spanning tree has the same number of edges. Then the edges in $\mathcal{O}_{\mathcal{S}}$ (where $\mathcal{S}$ is the target spanning tree we select) are the edges with smallest mean cost under parameter $\boldsymbol{\mu} \odot \mathcal{O}_{\mathcal{S}}$, since the other edges are set to have cost 1. Thus, any other spanning tree will induce a larger cost. As for the cascade bandit, the selected items have the highest click probabilities under the mean vector $\boldsymbol{\mu} \odot \mathcal{O}_{\mathcal{S}}$ for super arm $\mathcal{S}$, and replacing them to any other item will cause the total click probability to drop.

$(\alpha, \beta)$**-oracle**    Different from the previous CMAB literature, we do not consider the $(\alpha, \beta)$-approximation oracle and the $(\alpha, \beta)$-regret (Chen et al., 2016) in this paper. The fundamental problem for algorithms with $(\alpha, \beta)$-regret is that the algorithms are not "no-regret" algorithms, since they are not comparing to the best reward opt$(\mu)$, but $\alpha \cdot \beta \cdot$ opt$(\mu)$. Then, it is always possible to change an $(\alpha, \beta)$-oracle to $(\alpha, \beta - \epsilon)$-oracle by applying the original oracle with probability $1 - \epsilon$ and do the random exploration with probability $\epsilon$, which would make the problem unattackable. Thus, one should not hope to get general characterizations or results of the attackability for CMAB instances, and should discuss the attackability issue for a specific problem solved by specific algorithm and oracles. In the following, we show the attackability for the probabilistic maximum coverage problem solved by CUCB together with the Greedy oracle.

**Theorem 3.9.** *In the probabilistic maximum coverage problem (PMC), CUCB algorithm with Greedy oracle is polynomial attackable when $\Delta_{\mathcal{M}} > 0$. Besides, for any PMC instance, $\Delta_{\mathcal{M}} \geq 0$.*

The intuition of the proof is that although the Greedy oracle is an approximation oracle, by using CUCB, it "acts" like an exact oracle when the number of observations for each base arm is large enough, and thus we can follow the proof idea for Theorem 3.6.

Another interesting application of Theorem 3.6 is simple episodic reinforcement learning (RL) settings where the transition probability is known, i.e., white-box attack (Rakhsha et al., 2020; Zhang et al., 2020). Since the adversarial attacks on RL is a very large topic and little diverted from the current paper, we only provide minimal details for this result. Please refer to Appendix B.3 for more details.

**Corollary 3.10** (Informal)**.** *For episodic reinforcement learning setting where the transition probability is known, the reward poisoning attack for the episodic RL can be reduced to the adversarial attack on CMAB, and thus the "attackability" of the episodic RL instances are captured by Theorem 3.6.*

Although the above corollary is a simple *direct* application of Theorem 3.6 to white-box episodic RL, we anticipate that comparable techniques can be applied to analyze much more complex RL problems and provide an attackability characterization at the "instance" level. This part is left as future work.

## 4    ATTACK IN UNKNOWN ENVIRONMENT

In previous section, we discussed polynomial attackability of a CMAB instance in a known environment, i.e., all parameters of the instance such as the reward distributions of super arms and outcome distributions of base arms are given. However, in practice the environment is unknown to the adversary. Previous studies of adversarial attack on MAB and linear bandits showed that attackability of an instance in a known or an unknown environment Liu & Shroff (2019); Wang et al. (2022) are the same. In this section, we show that the polynomial attackability can be *different* for the *same* instance between a known or an unknown environment.

**Theorem 4.1.** *There exists a CMAB instance satisfying Assumption 2.1 and 2.2 such that it is polynomially attackable given the parameter $\boldsymbol{\mu}$ (induced from the instance's base arms' joint distribution $\mathcal{D}$), but there exists no attack algorithm that can efficiently attack the instances for CUCB algorithm with unknown parameter $\boldsymbol{\mu}$.*

The following example construct hard instances that satisfying Theorem 4.1. We also illustrate the example with $n = 5$ in Figure 4.

**Example 4.2** (Hard example). *We construct the following CMAB instance $\mathcal{I}_i$. There are $2n$ base arms, $\{a_i\}_{i \in [n]}$ and $\{b_i\}_{i \in [n]}$, and each corresponds to a random variable ranged in $[0, 1]$. We have $\mu_{a_j} = 1 - 2\epsilon$ for all $j \neq i$ and $\mu_{a_i} = 1$, and $\mu_{b_j} = 1 - \epsilon$ for all $j \in [n]$ for some $\epsilon > 0$. Then we construct $n + 2$ super arms. $\mathcal{S}_j$ for all $j \in [n]$ will observe base arms $a_j$ and $b_j$, and $r_{\mathcal{S}_j} = \mu_{a_j} + \mu_{b_j}$. There is another super arm $\mathcal{S}_{n+1}$ that will observe base arms $b_j$ for all $j \in [n]$, and $r_{\mathcal{S}_{n+1}} = \sum_{j \in [n]} \mu_{b_j} + (1 - \epsilon)$. Besides, there is also a super arm $\mathcal{S}_0$ with constant reward $\mathcal{S}_0 = 2 - 2\epsilon$ and does not observe any base arm. Then, the attack super arm set $\mathcal{M} = \{\mathcal{S}_j\}_{j \in [n]}$. In total, we can construct $n$ hard instances $\mathcal{I}_i$ for all $i \in [n]$.*

The intuition of why this example is hard is that it "blocks" the exploration of base arms $a_1, \ldots, a_m$ simultaneously. This means that the algorithm needs to visit exponential super arms and the cost is also exponential, violating the definition of polynomial attackable. Next, we give a proof sketch of Theorem 4.1 and explain more about the intuition, and the detailed proof is deferred to Appendix C.

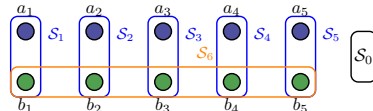

Figure 1: Example 4.2 with $n = 5$

*Proof sketch of Theorem 4.1.* First, given an instance $\mathcal{I}_i$, if we know the environment parameters $\boldsymbol{\mu}$, we can attack the CMAB instance by setting the rewards besides base arms $a_i, b_i$ to 0. However in the unknown setting, if we still want to attack the instance, we need to know the base arm $a_j$ for which $\mu_{a_j} = 1$. Thus, the attack algorithm needs to pull at least $\Omega(n)$ super arms in $\mathcal{M} = \{\mathcal{S}_j\}_{j \in [n]}$.

For simplicity, assume that the attack algorithm lets CUCB pull super arms $\mathcal{S}_1, \mathcal{S}_2 \ldots, \mathcal{S}_{n'}$ in order. Now if CUCB pulls super arm $\mathcal{S}_i$, it means that UCB $(a_i) \geq 1 - 2\epsilon$ and UCB $(b_i) \geq 1 - 2\epsilon$. Otherwise, UCB $(a_i)$ + UCB $(b_i) < 2 - 2\epsilon$ and CUCB will choose to play $\mathcal{S}_0$ instead. Besides, we also require UCB $(b_j) \leq \epsilon$ for all $j \neq i$. Otherwise, UCB $(b_i)$ + UCB $(b_j) + 1 - \epsilon >$ UCB $(a_i)$ + UCB $(b_i)$, and CUCB will choose $\mathcal{S}_{n+1}$.

Suppose that CUCB first pulls $\mathcal{S}_1$, and then pulls $\mathcal{S}_2$. When pulling $\mathcal{S}_1$, we know that UCB $(b_2) \leq \epsilon$, but if it needs to pull $\mathcal{S}_2$, UCB $(b_2) \geq 1 - 2\epsilon$. Note that $b_2$ can only be observed by $\mathcal{S}_2$ and $\mathcal{S}_{n+1}$, and without pulling $\mathcal{S}_2$, the attack algorithm needs to let CUCB to pull $\mathcal{S}_{n+1}$ to change the UCB value of $b_2$. Then CUCB needs to pull $\mathcal{S}_{n+1}$ for at least $\Omega(1/\epsilon)$ times. Now say the attack algorithm needs CUCB to pull $\mathcal{S}_3$, the UCB value of $b_3$ needs to raise to at least $1 - 2\epsilon$ from at most $\epsilon$. However, note that $b_3$ has already been observed by at least $\Omega(1/\epsilon)$ times since $\mathcal{S}_{n+1}$ is pulled by at least $\Omega(1/\epsilon)$ to observe $\mathcal{S}_2$. Then, $\mathcal{S}_{n+1}$ need to be pulled by $\Omega(1/\epsilon^2)$ times. Now if the attack algorithm wants CUCB to pull $\mathcal{S}_4$, CUCB will pull $\mathcal{S}_{n+1}$ for $\Omega(1/\epsilon^3)$ times. Because the attack algorithm needs CUCB to visit $\Omega(n)$ super arms in $\mathcal{M}$, the total cost is at least $1/\epsilon^{\Omega(n)} = 1/\epsilon^{\Omega(m)}$ since $m = 2n$. $\qquad\square$

Despite the hardness result mentioned previously, an adversary can still attack the CMAB instance using heuristics. For example, an adversary can just randomly pick a super arm $\mathcal{S}$ from the set $\mathcal{M}$ and set $\mathcal{S}$ to be the target arm in Algorithm 1. The guarantee of this heuristic follows from Theorem 3.6.

**Corollary 4.3.** *If $\mathcal{S}$ satisfies $\Delta_{\mathcal{S}} > 0$, then running Algorithm 1 can successfully attack the CMAB instance under Definition 3.1.*

Note that we have $\Delta_{\mathcal{M}} \geq 0$ for cascading bandit and online minimum spanning tree problem. Thus even in the unknown environment case, cascading bandit and online minimum spanning tree are mostly polynomially attackable (by applying Algorithm 1). Also from Theorem 3.9, when solving probabilistic covering problem using CUCB algorithm with Greedy oracle, $\Delta_{\mathcal{M}} \geq 0$ and thus it is also mostly polynomially attackable. However, for the shortest path problem, we do not know its attackability in the unknown environment since $\Delta_{\mathcal{M}}$ can be either positive or negative. Designing attacking algorithms for certain CMAB instances in the unknown setting is left as a future work.

## 5 NUMERICAL EXPERIMENTS

In this section, we present the numerical experiments along with their corresponding results. We empirically evaluate our attack on four CMAB applications: probabilistic maximum coverage, online minimum spanning tree , online shortest path problems , and cascading bandits . Additionally, we conduct experiments on the influence maximization problems discussed in Appendix A.3.

## 5.1 EXPERIMENT SETUP

**General setup** In our study, we utilize the Flickr dataset (McAuley & Leskovec, 2012) for the probabilistic maximum coverage, online minimum spanning tree, and online shortest path problems. We downsample a subgraph from this dataset and retain only the maximum weakly connected component to ensure connectivity, which comprises 1,892 nodes and 7,052 directed edges. We incorporate the corresponding edge activation weights provided in the dataset. For cascading bandits, we employ the MovieLens small dataset (Harper & Konstan, 2015), which comprises 9,000 movies. From this dataset, we randomly sample 5,000 movies for our experiments. Each experiment is conducted a minimum of 10 times, and we report the average results along with their variances. For a detailed setup, please refer to Appendix A.1.

**Probabilistic maximum coverage** We use the CUCB algorithm (Chen et al., 2016) along with a greedy oracle. We consider two kinds of targets: In the first case, we calculate the average weight of all outgoing edges of a node, sort the nodes with decreasing average weight, and select the nodes $K + 1, \ldots, 2K$, and the selected node set is denoted *fixed target*. The second type is the *random target*, where we randomly sample $K$ nodes whose average weight over all the outgoing edges is greater than $0.5$ to ensure no node with extremely sparse edge activation is selected.

**Online minimum spanning tree** We convert the Flickr dataset to an undirected graph, where use the average probability in both directions as the expected cost of the undirected edge. We employ a modified version of Algorithm 1, with line 4 changed to $\tilde{X}_i^T = 1$ since now we are minimizing the cost instead of maximizing the reward. We consider two types of targets: (1) the *fixed target*, where $\mathcal{M}$ containing only the second-best minimum spanning tree; and (2) the *random target*, where $\mathcal{M}$ contains the minimum spanning tree obtained on the graph with same topology but *randomized weight* (mean uniformly sampled from $[0, 1]$) on edges.

**Online shortest path** We consider two types of targets $\mathcal{M}$: In the first type, $\mathcal{M}$ contains an unattackable path that is carefully constructed. We randomly draw a source node $\mathbf{s}$, then use random walk to unobserved nodes and record the path from $\mathbf{s}$. When we reach a node $k$ and the path weight is larger than the shortest path length from $\mathbf{s}$ to $k$ by a certain threshold $\theta$, we set $k$ as the destination node $\mathbf{t}$ and set the path obtained from the random walk as the target path. If after $50$ steps the target path still can not be found, we re-sample the source node $\mathbf{s}$. To avoid trivial unattackable cases, we require the shortest path should have more than one edge. In our experiment, we set the threshold $\theta$ as $0.5$. We call this type of $\mathcal{M}$ as *unattackable target*. The second type of $\mathcal{M}$ is generated by randomly sampling the source node $\mathbf{s}$ and the destination node $\mathbf{t}$, and then finding the shortest path from $\mathbf{s}$ to $\mathbf{t}$ given a randomized weight on edges. We call this type of $\mathcal{M}$ as *random target*. We sample 100 targets for both *unattackable* and *random* ones, and repeat the experiment 10 times for each target.

**Cascading bandit** In this experiment, we test CascadeKL-UCB, CascadeUCB1 (Kveton et al., 2015), and CascadeUCB-V (Vial et al., 2022). We compute a $d$-rank SVD approximation using the training data, which is used to compute a mapping $\phi$ from movie rating to the probability that a user selected at random would rate the movie with 3 stars or above. For further details, we direct the reader to Section 6 and Appendix C of Vial et al. (2022). In our experiments, we select the target super arm from the subset of a base arms $\mathcal{M}$ whose average click probability is greater than $0.1$.

## 5.2 EXPERIMENT RESULTS

The experiment results are summarized in Figure 2, which show the cost and number of target super arm played under different settings.

**Probabilistic maximum coverage** Fig. 2(a), 2(b) show the results of our experiments on two types of targets. From Fig. 2(b) and 2(a), we observe that the number of target arm pulls increases linearly after around $5,000$ iterations while the cost grows sublinearly. Considering the large number of nodes and edges, the cost is clearly polynomial to the number of base arms. This result validated our Theorem 3.9 that probabilistic maximum covering is polynomially attackable. We also report the variance in the figures, highlighting that attacking a random target results in larger variances for both cost and target arm pulls.

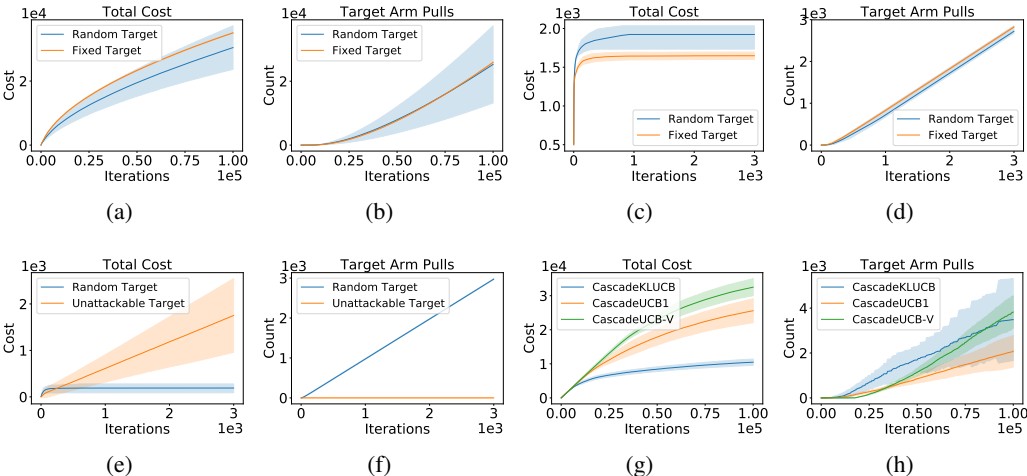

Figure 2: Cost and target arm pulls for: (2(a), 2(b)) probabilistic max coverage; (2(c), 2(d)) online maximum spanning tree; (2(e), 2(f)) online shortest path; (2(g), 2(h)) cascading bandits. Experiments are repeated for at least 10 times and we report the averaged result and its variance.

**Online minimum spanning tree**    As demonstrated in Figs 2(c) and 2(d), the total cost is sublinear and the rounds pulling the target arm are linear. This result aligns with our Corollary 3.8: since the number of edges in spanning tree is the same and the 'reward' of the super arm is a linear combination of base arms, $\Delta_{\mathcal{M}} \geq 0$. Thus online minimum spanning tree problem is always attackable as suggested by experiments on a random target if there is only one minimum spanning tree.

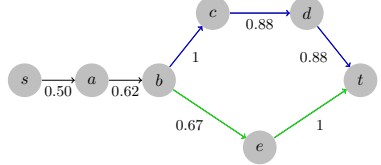

Figure 3: An unattackable shortest path from $s$ to $t$ in the Flickr dataset. Optimal path: $(s, a, b, e, t)$. Target path: $(s, a, v, c, d, t)$. The cost on $(b, c, d, t)$ is larger than the number of edges on $(b, e, t)$, and the attacker cannot fool the algorithm to play the target path.

**Online shortest path**    In Figs 2(e) and 2(f), we show that for the random target $\mathcal{M}$, the total cost is sublinear, and the target arm pulls are linear. For the unattackable targets $\mathcal{M}$, the total cost is linear while the target arm pulls are almost constant, which means they are indeed unattackable. We show an example from the Flickr dataset in Figure 3.

**Cascading bandits**    We can clearly observe from Fig. 2(h) & 2(g) the number of times the target arm is played increases linearly, while the cost of attack is sublinear. Considering the large number of base arms $m$, the results validated our Corollary 3.8 that cascading bandits is polynomially attackable.

## 6    CONCLUSION

In this paper, we provide the first study on attackability of CMAB and propose to consider the *polynomial attackability*, a definition accommodating the exponentially large number of super arms in CMAB. We first provide a sufficient and necessary condition of polynomial attackability of CMAB. Our analysis reveals a surprising fact that the polynomial attackability of a particular CMAB instance is conditioned on whether the bandit instance is known or unknown to the adversary. In addition, we present a hard example, such that the instance is polynomially attackable if the environment is known to the adversary and polynomially unattackable if it is unknown. Our result reveals that adversarial attacks on CMAB are difficult in practice, and a general attack strategy for any CMAB instance does not exist since the environment is mostly unknown to the adversary. We propose an attack algorithm where we show that the attack is guaranteed to be successful and efficient if the instance is attackable. Extensive experimental results on real-world datasets from four applications of CMAB validate the effectiveness of our attack algorithm. As of future work, we will further investigate the impact of probabilistic triggered arm to the attackability, e.g., the attackability of online influence maximization given different diffusion models. It is also important to generalize the hardness result in unknown environment, e.g., a general framework to reduce unattackable CMAB problems to the hardness example.

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

APPENDIX

# A  ADDITIONAL EXPERIMENTS

In this section, we provide additional information regarding the experiments. In Appendix A.1, we give more details for the experiments in Section 5. In Appendix A.3, we show the results of our experiments on online influence maximization.

## A.1  MORE EXPERIMENT DETAILS

For the online shortest path, online minimum spanning tree and probabilistic maximum coverage we use the Flickr dataset (McAuley & Leskovec, 2012). We first select a fraction of the nodes such that their degree is in the range $[m, n]$. Here, we take $m = 2, n = 75$. We then clean up the graph by selecting the largest connected component, which has $1892$ nodes and $7052$ edges.

For cascading bandits, we use the small version of the MovieLens Dataset (Harper & Konstan, 2015), which contains $9000$ movies. Each movie has a rating (in the range of one to five stars) given to it by several users. To convert the ratings from stars to average click probability by any user, we consider the click probability close to the probability that the movie has a rating of $3$ stars or above. We follow Vial et al. (2022) to generate a mapping $\phi$ that goes from ratings in stars to the average click probability for some user selected at random. However, as the average click probabilities are sparse, we use only the top $5000$ movies in our experiments.

## A.2  ADDITIONAL RESULTS ON CASCADING BANDITS

We provide additional results for the Cascading Bandits setting here. Fig. 4(a), 4(b), show the cost and trigger rate of when selecting 25 items out of 5000 ($K = 25, m = 5000$). However, we can see that the total number of times the target super arm $S$ is played is much lesser than the number of iterations. To assert that our algorithm is still capable of efficiently attacking cascading bandits, we run additional experiments with $K = 25, m = 1000$ (Fig. 4(c), 4(d)) and $K = 5, m = 1000$ (Fig. 4(e), 4(f)). We clearly observe that as we simplify the problem, the target arm trigger rate substantially increases, and the slope of the curves nears $1$, showing that the target set $S$ is selected linear number of times. Furthermore, the sublinearity of the cost is more apparent from these experiments. This substantiates our claim that our attack algorithm successfully attacks the cascading bandit environment, and with more compute and iterations/time, it is possible to attack the scenario with 5000 and 9000 items.

## A.3  EXPERIMENT RESULTS ON INFLUENCE MAXIMIZATION

Here we provide additional experiment results for the online influence maximization (IM) problem.

**Basic influence maximization settings.**  The IM problem involves selecting (activating) an initial set of $K$ nodes. Each node $u$ can attempt to activate its neighbor $v$ with probability $p_{u,v}$. If node $v$ was not activated by node $u$ in time instant $i$, it cannot be activated by $u$ in any further time instant $i + 1, i + 2, \cdots$. This is called one step of diffusion. We continue this process until time step $T_0$, such that no new node can be activated in time step $T_0 + 1$. The goal of the IM problem is to select a set of $K$ nodes that maximize the final number of activated nodes. The goal in online IM is to select a set of $K$ nodes at each round $t$ that minimizes the total regret. At each round, the player observes the diffusion process, and receives the number of activated nodes as the reward. Note that the IM problem is very similar to the probabilistic maximum coverage problem, where the only difference is the length of the diffusion process since the probabilistic maximum coverage problem can be viewed as influence maximization when there is only one diffusion step.

**Attack heuristics.**  Note that the IM problem is NP-hard and only has $(1 - 1/e)$-approximation oracle. Thus, we do not have an attack strategy that has a theoretical guarantee from Theorem 3.6. However, we can still have heuristics to attack the online influence maximization problem.

We define a set $S_{ex}^{\ell}$ that represents the extended target set. This set includes the target nodes and all other nodes $v$ at a distance of at most $\ell$ from any node $u$ in the target set $S$,

$$S' = \{v : \exists u \in S, v \in V, d(u, v) < \ell\}, \quad S_{ex}^{\ell} = S \cup S'$$

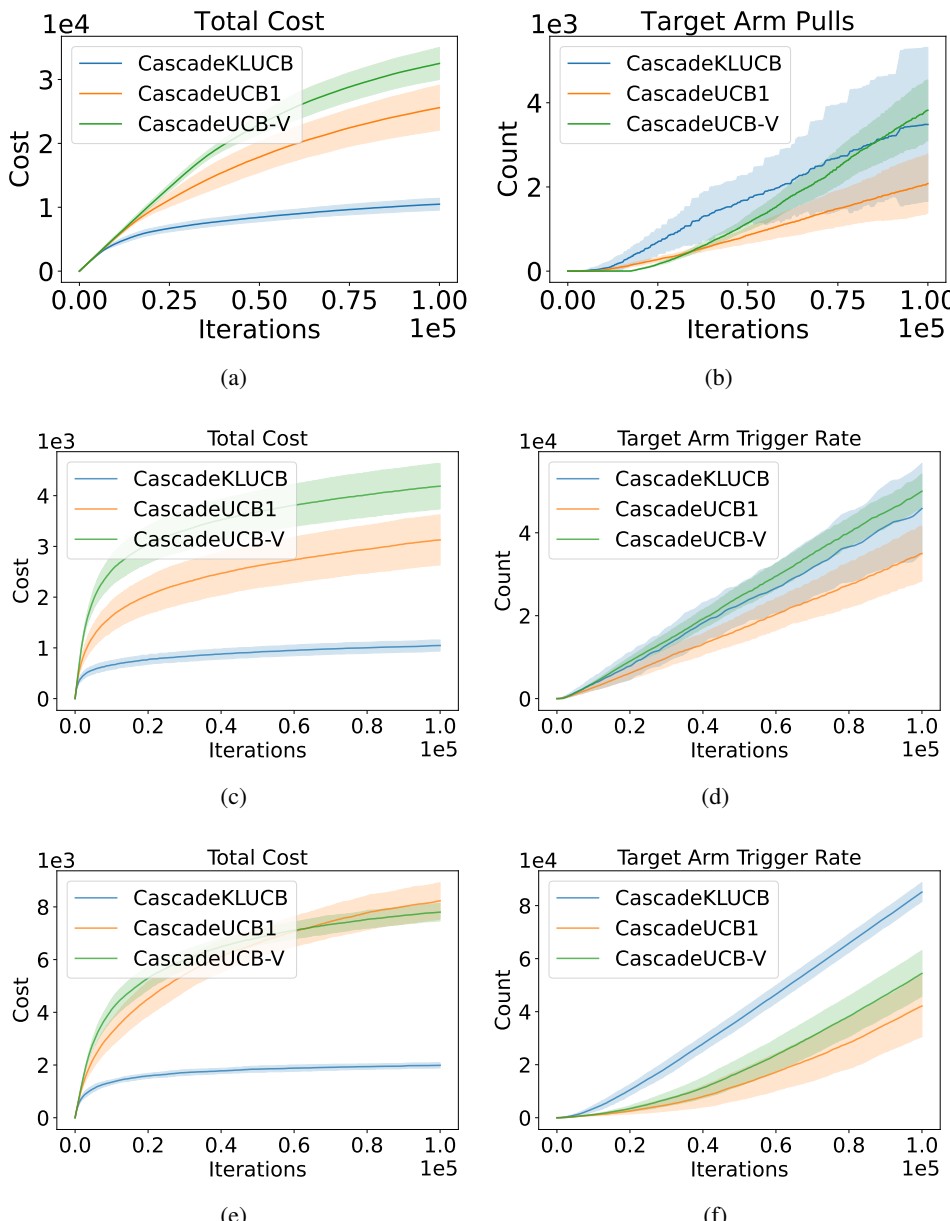

Figure 4: Cost and target arm pulls for cascading bandits on the MovieLens-small dataset with (4(a), 4(b)) $m = 25, K = 5000$, (4(c), 4(d)) $m = 35, K = 1000$, (4(e), 4(f)) $m = 5, K = 1000$. Experiments are repeated for at least 10 times and we report the averaged result and its variance.

where $d(u, v)$ is the distance function between two nodes $u$ and $v$. We do not attack the edge between $u$ and $v$ if either of the nodes lies in $S_{ex}^{\ell}$. For all other observed edges, we modify the realization to 0 (thus attack it). Note that if $\ell = \infty$, then $S_{ex}^{\ell}$ contains all nodes, and thus there is no viable attack. If $\ell = 1$, then $S_{ex}^{\ell}$ only includes the target set, and then the reward feedback is very similar to the probabilistic maximum coverage problem, and the "expected value after attack" for each arm is exactly the same as the probabilistic maximum coverage attack strategy (Theorem 3.9). This attack heuristic simplifies the strategy for the attacker as there are more nodes in the extended set than in the target set, making the attack easier. In all the experiments, we use an $(\alpha, \beta)$-approximation oracle based on (Tang et al., 2015) (IMM oracle) and Fig 5 shows the corresponding results. Experiments are repeated for at least 5 times and we report the averaged result and its variance.

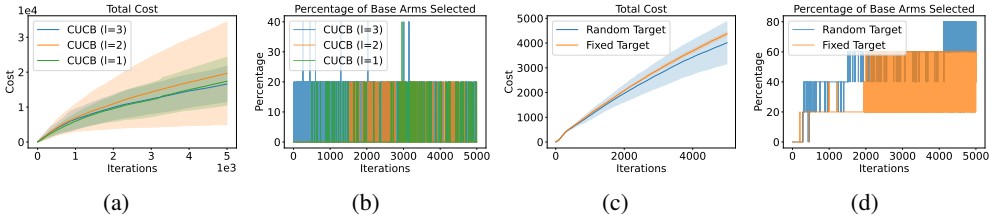

Figure 5: Cost and percentage of base arms selected for: (5(a), 5(b)) Influence Maximization; (5(c), 5(d)) Probabilistic Maximum Coverage.

**Discussions of the results.**  Figure 5(b) and Figure 5(d) show the percentage of target nodes played in each round for online influence maximization and probabilistic maximum coverage respectively. This value represents the percentage of base arms in the target super arm that is selected at a given time instant. For example, if we consider a target set containing $5$ nodes (base arms), $20\%$ of base arms selected would imply that at that round, $1$ node belonging to the target set was played.

From Figure 5, we can clearly find that although the attack cost in each round decreases as the number of iterations increases[2], the number of target set played is a constant $0$. This may happen because (1) attacking online influence maximization is hard, and (2) the number of iterations is not large enough. To prove the attacking influence maximization is harder, we also show the result of probabilistic maximum coverage only with $T = 5000$ (Figure 5). Although for probabilistic maximum coverage, there is also no target set pulled in the first 5000 rounds, we can clearly observe that the percentage of target nodes selected is increasing, and there is a clear trend to select all the nodes in the target arm set when the number of iteration is large enough (which is the case in Figure 2(b)). However, for the online influence maximization problem, the algorithm selects none or 20% of the target node set for $\ell = 1, 2, 3$ for a majority of the experiment, and there is no trend indicating that the number of target nodes selected would increase. Note that when $\ell = 1$, the reward structure of influence maximization and probabilistic maximum coverage are nearly the same, and the difference is the oracle. This finding corroborates our claim that when the oracle for a CMAB instance is not exact ($\alpha$-approximation oracle), we need to analyze the instance case by case.

Although we do not have a promising attack strategy for online influence maximization, it does not mean that it is unattackable. Further developing efficient attack methods for online influence maximization problem or proving its intrinsic robustness is a very interesting future work.

## B  Missing proofs in Section 3

In this section, we give the missing proofs in Section 3. We first prove Theorem 3.6 in Appendix B.1, which characterizes the "attackability" issue of different CMAB instances under the exact oracle. Then in Appendix B.2, we prove Theorem 3.9, which has an approximation oracle.

### B.1  Proof of Theorem 3.6

In this section, we prove Theorem 3.6, which is restated below.

**Theorem 3.6** (Polynomial attackability of CMAB). *Given a particular CMAB instance and the target set of super arms $\mathcal{M}$ to attack. If $\Delta_{\mathcal{M}} > 0$, then the CMAB instance is polynomially attackable. If $\Delta_{\mathcal{M}} < 0$, the instance is polynomially unattackalble.*

The proof of Theorem 3.6 is divided into two cases: $\Delta_{\mathcal{M}} > 0$ or $\Delta_{\mathcal{M}} < 0$. We first show the case when $\Delta_{\mathcal{M}} > 0$, and then we prove the theorem when $\Delta_{\mathcal{M}} < 0$.

---

[2]Here in theory, in our attack strategy, the attack cost can never be sublinear, since no matter what $l$ parameter we choose, there is a "constant" probability (independent on the time $t$) that the diffusion happens to the nodes outside $S_{ex}^l$. However, if we choose $l$ large enough, the probability to attack will decrease, and if we choose the hyperparameter $l$ appropriately to make sure the probability to attack is $o(1)$, the attack cost can still be "sublinear" with respect to a given time scale $T$.

**Proof of Theorem 3.6 when $\Delta_{\mathcal{M}} > 0$.** The proof of this direction is relatively straightforward: we only need to show that there exists an attacking strategy that successfully attacks the CMAB instance when the number of rounds $T$ is polynomially large. The attacking strategy is exactly Algorithm 1.

Note that if $\Delta_{\mathcal{M}} > 0$, we can find $\mathcal{S}^*$ such that $\Delta_{\mathcal{S}^*} > 0$. The regret of the CMAB algorithm Alg is bounded by $\text{poly}(m, 1/p^*, K)T^{1-\gamma}$ with high probability, then in the modified CMAB environment with $\boldsymbol{\mu}' = \boldsymbol{\mu} \odot \mathcal{O}_{\mathcal{S}^*}$, Alg will only play super arms other than $\mathcal{S}^*$ for $\text{poly}(m, 1/p^*, K)T^{1-\gamma}/\Delta_{\mathcal{S}^*}$ times. Each time when Alg pulls super arm other than $\mathcal{S}^*$, the attacking algorithm (Algorithm 1) might need to modify the reward, leading to the cost bounded by $m$. Therefore the total budget can be bounded by $\text{poly}(m, 1/p^*, K) \cdot m \cdot (1/\Delta_{\mathcal{S}^*}) \cdot T^{1-\gamma}$. We conclude the proof by choosing $T^{\gamma/2} \geq \text{poly}(m, 1/p^*, K) \cdot m \cdot (1/\Delta_{\mathcal{S}^*})$ and set $\gamma' = \gamma/2$.

**Proof of Theorem 3.6 when $\Delta_{\mathcal{M}} < 0$.** Now we prove the other direction: when $\Delta_{\mathcal{M}} < 0$, the CMAB instance is polynomially unattackable (Definition 3.2). The intuition of the proof is that: when all base arms $i \in \cup_{\mathcal{S} \in \mathcal{M}} \mathcal{O}_{\mathcal{S}}$ are observed for a certain number of times (say, $T^{1-\gamma'/2}$ times for each base arm), CUCB will not choose to play the super arms in $\mathcal{M}$. Since the budget is bounded by $\mathcal{B} \leq T^{1-\gamma'}$, the estimated upper confidence bound of CUCB after corrupted by the attacker should be still close to the real empirical mean. Thus, because $\Delta_{\mathcal{M}} < 0$, the CUCB algorithm will not try to play the arms in $\mathcal{M}$ anymore. The following part formalizes this intuition. The following proof is organized as follows: we first present the necessary notations, technical propositions, definitions, and lemmas; then, we formalize the intuition in the main lemma (Lemma B.8); finally, we conclude the proof base on Lemma B.8.

We use $\tilde{\mu}_i^{(t)}$ to denote the mean of base arm $i$ estimated by the CUCB algorithm (possibly after attacker's corruption). We use $T_i^{(t)}$ to denote the number of times base arm $i$ is observed before round $t$. We use $\rho_i^{(t)}$ to denote the confidence bound of base arm $i$. Because we are interested in a high-probability event, we choose $\rho_i^{(t)} = \sqrt{\frac{\ln(4mt^3/\delta)}{2T_{i,t}}}$. We also define $\hat{\mu}_{i,t}$ to be the empirical mean generated by the environment before the corruption by the attacker (or the empirical mean observed by the attacker from the environment). We define $\mathbf{UCB}_t = \min\{\tilde{\boldsymbol{\mu}}^{(t)} + \boldsymbol{\rho}^{(t)}, 1\}$ to be the upper confident bound at round $t$.

First, we show two standard concentration inequalities (Proposition B.1 and B.2) and several technical results.

**Proposition B.1** (Hoeffding Inequality). *Suppose $X_i \in [0, 1]$ for all $i \in [n]$ and $X_i$ are independent, then we have*

$$\Pr\left\{\left|\frac{1}{n}\sum_{i=1}^{n} X_i - \mathbb{E}\left[\frac{1}{n}\sum_{i=1}^{n} X_i\right]\right| \geq \epsilon\right\} \leq 2\exp\left(-2n\epsilon^2\right).$$

**Proposition B.2** (Multiplicative Chernoff Bound). *Suppose $X_i$ are Bernoulli variables for all $i \in [n]$ and $\mathbb{E}[X_i|X_1, \ldots, X_{i-1}] \geq \mu$ for every $i \leq n$. Let $Y = X_1 + \cdots + X_n$, then we have*

$$\Pr\{Y \leq (1-\delta)n\mu\} \leq \exp\left(-\frac{\delta^2 n\mu}{2}\right).$$

Next, we define the following event that the empirical means of different base arms returned by the environment is close to the mean $\boldsymbol{\mu}$. This definition is similar to that in Wang & Chen (2017).

**Definition B.3** (Sampling is Nice). We say that the sampling is nice at the beginning of round $t$ if for any arm $i \in [m]$, we have $|\hat{\mu}_i^{(t)} - \mu_i^{(t)}| < \rho_i^{(t)}$, where $\rho_i^{(t)} = \sqrt{\frac{\ln(4mt^3/\delta)}{2T_i^{(t)}}}$ ($\infty$ if $T_i^{(t)} = 0$). We use $\mathcal{N}_t^s$ to denote this event.

The following lemma shows that the defined event should hold with high probability. The proof is a straightforward application of Hoeffding's inequality (Proposition B.1).

**Lemma B.4.** *For each round $t \geq 1$, we have $\Pr\{\neg\mathcal{N}_t^s\} \leq \frac{\delta}{2t^2}$. Besides, $\Pr\{\cup_t(\neg\mathcal{N}_t^s)\} \leq \delta$.*

*Proof.* For each round $t \geq 1$, we have

$$\Pr\{\neg\mathcal{N}_t^s\} = \Pr\left\{\exists i \in [m], |\hat{\mu}_i^{(t)} - \mu_i| \geq \sqrt{\frac{\ln(4mt^3/\delta)}{2T_i^{(t)}}}\right\}$$

$$\leq \sum_{i\in[m]} \Pr\left\{|\hat{\mu}_i^{(t)} - \mu_i| \geq \sqrt{\frac{\ln(4mt^3/\delta)}{2T_i^{(t)}}}\right\}$$

$$= \sum_{i\in[m]} \sum_{k=1}^{t-1} \Pr\left\{T_i^{(t)} = k, |\hat{\mu}_i^{(t)} - \mu_i| \geq \sqrt{\frac{\ln(4mt^3/\delta)}{2T_i^{(t)}}}\right\}.$$

Then we apply Hoeffding's inequality, and we have

$$\Pr\left\{T_i^{(t)} = k, |\hat{\mu}_i^{(t)} - \mu_i| \geq \sqrt{\frac{\ln(4mt^3/\delta)}{2T_i^{(t)}}}\right\} \leq 2e^{-2k\frac{\ln(4mt^3/\delta)}{2k}} = \frac{\delta}{2mt^3}.$$

Plugging in the previous bound, we have $\Pr\{\neg\mathcal{N}_t^s\} \leq \frac{\delta}{2t^2}$. Now we take the union bound over $t$, we have

$$\Pr\{\cup_t(\neg\mathcal{N}_t^s)\} \leq \sum_{t=1}^{\infty} \frac{\delta}{2t^2} \leq \delta,$$

and we conclude the proof. $\qquad\square$

Because of the probabilistic triggered arms, a base arm may only be observed with some probability when some super arm is pulled. For the sake of simpler analysis, we define the following notion "Counter", which characterizes how many times a base arm may be triggered by some super arm.

**Definition B.5** (Counter). In an execution of the CUCB algorithm, we define the counter $N_{i,t}$ as the following number

$$N_{i,t} := \sum_{s=0}^{t} \mathbb{I}\left\{p_i^{\mathcal{D},\mathcal{S}_s} > 0\right\}.$$

The following event (Definition B.6) bridges the observation time of a base arm and the notion "Counter". Then, Lemma B.7 shows that the following event should hold with high probability. The proof of Lemma B.7 is a direct application of the multiplicative Chernoff bound (Proposition B.2).

**Definition B.6** (Triggering is Nice). We call that the triggering is nice at the beginning of round $t$ if for any arm $i$, as long as $\ln(4mt^3/\delta) \leq \frac{1}{4}N_{i,t-1} \cdot p^*$, we have

$$T_i^{(t-1)} \geq \frac{1}{4}N_{i,t-1} \cdot p^*.$$

We use $\mathcal{N}_t^t$ to denote this event.

**Lemma B.7.** *We have for every round $t \geq 1$,*

$$\Pr\{\neg\mathcal{N}_t^t\} \leq \frac{\delta}{4t^3}.$$

*Then, we have*

$$\Pr\{\cup(\neg\mathcal{N}_t^t)\} \leq \frac{\delta}{2}.$$

*Proof.* We directly apply the Multiplicative Chernoff Bound (Proposition B.2). Note that as long as $N_{i,t-1} \cdot p^* \geq 4\ln(4mt^3/\delta)$, we have

$$\Pr\left\{T_i^{(t-1)} \leq \frac{1}{4}N_{i,t-1} \cdot p^*\right\} \leq \exp\left(-\frac{3^2 N_{i,t-1} \cdot p^*}{2 \times 4^2}\right)$$

$$\leq \exp\left(-\frac{9 \times 4\ln(4mt^3/\delta)}{32}\right)$$

$$\leq \exp\left(\ln(4mt^3/\delta)\right)$$

$$\leq \frac{\delta}{4mt^3}.$$

Now applying the union bound on $i \in [m]$ and $N_{i,t-1}$, we can get

$$\Pr\{\neg\mathcal{N}_t^t\} \leq \frac{\delta}{4t^2}.$$

Then apply the union bound on $t$, we have

$$\Pr\{\cup(\neg\mathcal{N}_t^t)\} \leq \sum_{t \geq 1} \frac{\delta}{4t^2} \leq \frac{\delta}{2} \frac{\pi^2}{6 \times 2} \leq \frac{\delta}{2}.$$

□

Given the previous technical lemmas, we now show the main lemma to prove Theorem 3.6 when $\Delta_\mathcal{M} < 0$. The following lemma upper bounds the number of times the CUCB algorithm plays the arms in $\mathcal{M}$ under certain budget constraint $\mathcal{B}$.

**Lemma B.8.** *Suppose that the adversary only has budget $\mathcal{B} = o(T)$ and assume that $\cap\mathcal{N}_t^s$ holds. For any $\mathcal{S} \in \mathcal{M}$, if $\Delta_\mathcal{M} < 0$, and $\mathcal{S}$ is pulled in round $t$, then there must exist $i \in \mathcal{O}_\mathcal{S}$ such that*

$$T_i^{(t)} \leq T_0 := \frac{6KB\mathcal{B}}{|\Delta_\mathcal{M}|} + \frac{18K^2B^2\log(4mT^3/\delta)}{|\Delta_\mathcal{M}|^2}.$$

*Proof.* We'll use proof by contradiction. Let's suppose there exists a round $t$ where $\mathcal{S}$ is pulled and $T_i^{(t)} \geq T_0$ for all $i \in \mathcal{S}$. Since the adversary only has budget $\mathcal{B} = o(T)$, each arm $i$ can receive the corrupted value for at most $\mathcal{B}$ times. Thus, we have for all $i \in \mathcal{S}$,

$$|\tilde{\mu}_i^{(t)} - \hat{\mu}_i^{(t)}| \leq \frac{\mathcal{B}}{T_0} \leq \frac{|\Delta_\mathcal{S}|}{6KB},$$

where we use the fact that $|\Delta_\mathcal{S}| \geq |\Delta_\mathcal{M}|$. Besides, given $\cap\mathcal{N}_t^s$, for all $i \in \mathcal{S}$ we have

$$|\hat{\mu}_i^{(t)} - \mu_i| \leq \rho_i^{(t)} = \sqrt{\frac{\log(4mt^3/\delta)}{2T_i^{(t)}}} \leq \frac{|\Delta_\mathcal{S}|}{6KB}.$$

Then under $\cap\mathcal{N}_t^s$, we have

$$
\begin{aligned}
\mathbf{UCB}_t \odot \mathcal{O}_\mathcal{S} - \boldsymbol{\mu} \odot \mathcal{O}_\mathcal{S} &= ((\tilde{\boldsymbol{\mu}}^{(t)} + \boldsymbol{\rho}^{(t)}) \odot \mathcal{O}_\mathcal{S}) - \boldsymbol{\mu} \odot \mathcal{O}_\mathcal{S} \\
&= (\tilde{\boldsymbol{\mu}}^{(t)} + \boldsymbol{\rho}^{(t)} - \boldsymbol{\mu}) \odot \mathcal{O}_\mathcal{S} \\
&\preceq (\hat{\boldsymbol{\mu}}^{(t)} + \frac{|\Delta_\mathcal{S}|}{6KB}\mathbf{1} + \boldsymbol{\rho}^{(t)} - \boldsymbol{\mu}) \odot \mathcal{O}_\mathcal{S} \\
&\preceq (\boldsymbol{\mu} + \frac{|\Delta_\mathcal{S}|}{6KB}\mathbf{1} + 2\boldsymbol{\rho}^{(t)} - \boldsymbol{\mu}) \odot \mathcal{O}_\mathcal{S} \\
&\preceq (\frac{|\Delta_\mathcal{S}|}{6KB}\mathbf{1} + 2\frac{|\Delta_\mathcal{S}|}{6KB}\mathbf{1}) \odot \mathcal{O}_\mathcal{S} \\
&\preceq \frac{|\Delta_\mathcal{S}|}{2KB} \odot \mathcal{O}_\mathcal{S}.
\end{aligned}
$$

Let $\mathcal{S}'$ denote a super arm such that $r_{\mathcal{S}'}(\mu \odot \mathcal{O}_{\mathcal{S}}) = r_{\mathcal{S}}(\mu) - \Delta_{\mathcal{S}}$, under $\cap \mathcal{N}_t^s$, we have

$$
\begin{aligned}
r_{\mathcal{S}'}(\mathbf{UCB}^{(t)}) &= r_{\mathcal{S}'}(\tilde{\boldsymbol{\mu}}^{(t)} + \boldsymbol{\rho}^{(t)}) \\
&\geq r_{\mathcal{S}'}((\tilde{\boldsymbol{\mu}}^{(t)} + \boldsymbol{\rho}^{(t)}) \odot \mathcal{O}_{\mathcal{S}}) \\
&\geq r_{\mathcal{S}'}\left(\left(\hat{\boldsymbol{\mu}}^{(t)} - \frac{|\Delta_{\mathcal{S}}|}{6KB}\mathbf{1} + \boldsymbol{\rho}^{(t)}\right) \odot \mathcal{O}_{\mathcal{S}}\right) \\
&\geq r_{\mathcal{S}'}\left((\hat{\boldsymbol{\mu}}^{(t)} + \boldsymbol{\rho}^{(t)}) \odot \mathcal{O}_{\mathcal{S}}\right) - \sum_{i \in \mathcal{S}'} B\frac{|\Delta_{\mathcal{S}}|}{6KB} \\
&\geq r_{\mathcal{S}'}(\boldsymbol{\mu} \odot \mathcal{O}_{\mathcal{S}}) - \frac{|\Delta_{\mathcal{S}}|}{6} \\
&= r_{\mathcal{S}}(\boldsymbol{\mu} \odot \mathcal{O}_{\mathcal{S}}) + \frac{5}{6}|\Delta_{\mathcal{S}}| \\
&\geq r_{\mathcal{S}}(\mathbf{UCB}_t) - \sum_{i \in \mathcal{S}} B\frac{|\Delta_{\mathcal{S}}|}{2KB} + \frac{5}{6}|\Delta_{\mathcal{S}}| \\
&\geq r_{\mathcal{S}}(\mathbf{UCB}_t) + \frac{1}{3}|\Delta_{\mathcal{S}}|,
\end{aligned}
$$

which means that CUCB will not play $\mathcal{S}$, which contradicts our assumption that $\mathcal{S}$ is pulled in round $t$. Thus, our original proposition that there exists an element $i \in \mathcal{O}_{\mathcal{S}}$ such that $T_i^{(t)} \leq T_0$ holds true, completing the proof. $\qquad\square$

Then we are ready to prove Theorem 3.6 when $\Delta_{\mathcal{M}} < 0$.

*Proof of Theorem 3.6 when $\Delta_{\mathcal{M}} < 0$.* First we assume that $\cap \mathcal{N}_t^s$ and $\cap \mathcal{N}_t^t$ hold, which should happen with probability at least $1 - \frac{3}{2}\delta$.

Now under $\cap \mathcal{N}_t^s$, if $\mathcal{S} \in \mathcal{M}$ is pulled, then at least one of the base arm $i \in \mathcal{O}_{\mathcal{S}}$ has not been observed for $T_0$ times. Besides, under $\cap \mathcal{N}_t^t$, we know that when $\mathcal{S}$ is pulled at time $t$, at least one of its base arm $i \in \mathcal{O}_{\mathcal{S}}$ satisfies $N_{i,t-1} \leq 4T_0/p^*$.

Note that there are $m$ arms in total, thus the CUCB algorithm will only play super arms belonging to $\mathcal{M}$ for at most $4mT_0/p^*$ times in total. Since every time CUCB pulls $\mathcal{S} \in \mathcal{M}$, the quantity $\sum_i N_{i,t}$ increases by at least 1. Note that if $\mathcal{B} \leq T^{1-\gamma'}$, $T_0$ can be bounded by $\text{poly}(m, 1/p^*, K, 1/|\Delta_{\mathcal{M}}|, \log(1/\delta)) \cdot T^{1-\gamma'}$. Thus there must exist some polynomial on $T^* = \text{poly}(m, 1/p^*, K, 1/|\Delta_{\mathcal{M}}|, \log(1/\delta))$ such that when $T > T^*$, $4nT_0/p^* \leq \frac{T}{2}$, and thus we conclude the proof. $\qquad\square$

## B.2 Proof of Theorem 3.9

In this section, we prove Theorem 3.9. The intuition of Theorem 3.9 is that, although the Greedy oracle is an approximation oracle, by using CUCB, it "acts" like an exact oracle when the number of observations for each base arm is large enough, and thus we can follow the proof idea for Theorem 3.6. We first present some technical lemmas to leverage the submodular structure of the reward function.

**Lemma B.9** (Chen et al. (2013)). *Probabilistic maximum coverage is 1-TPM bounded smoothness (Assumption 2.2).*

**Lemma B.10.** *Suppose $\mathcal{S} = \{u_1, u_2, \ldots, u_k\}$ denote one super arm the probabilistic maximum coverage problem, then for any $u \in \mathcal{S}$ and any set $\mathcal{S}' \subseteq \mathcal{S}$ and $u \notin \mathcal{S}'$, we have,*

$$
r_{\mathcal{S}' \cup \{u\}}(\boldsymbol{\mu} \odot \mathcal{O}_{\mathcal{S}}) - r_{\mathcal{S}'}(\boldsymbol{\mu} \odot \mathcal{O}_{\mathcal{S}}) \geq \Delta_{\mathcal{S}}.
$$

*Besides, $\Delta_{\mathcal{S}} \geq 0$.*

*Proof.* First for any $u \in \mathcal{S}$ and $u' \notin \mathcal{S}$, we have

$$
r_{\mathcal{S}}(\boldsymbol{\mu} \odot \mathcal{O}_{\mathcal{S}}) - r_{(\mathcal{S} \setminus \{u\}) \cup \{u'\}}(\boldsymbol{\mu} \odot \mathcal{O}_{\mathcal{S}}) \geq \Delta_{\mathcal{S}}.
$$

Observe that $\mathcal{O}_\mathcal{S}$ will assign 0 to all edges without an endpoint in $\mathcal{S}$, and thus

$$r_{(\mathcal{S}\setminus\{u\})\cup\{u'\}}(\boldsymbol{\mu}\odot\mathcal{O}_\mathcal{S}) = r_{\mathcal{S}\setminus\{u\}}(\boldsymbol{\mu}\odot\mathcal{O}_\mathcal{S}).$$

Then because the reward function of the probabilistic maximum coverage problem is submodular in terms of the set $\mathcal{L}$ (Chen et al., 2013), we then have for any $\mathcal{S}'\subseteq\mathcal{S}$ such that $u\notin\mathcal{S}'$,

$$r_\mathcal{S}(\boldsymbol{\mu}\odot\mathcal{O}_\mathcal{S}) - r_{\mathcal{S}\setminus\{u\}}(\boldsymbol{\mu}\odot\mathcal{O}_\mathcal{S}) \le r_{\mathcal{S}'\cup\{u\}}(\boldsymbol{\mu}\odot\mathcal{O}_\mathcal{S}) - r_{\mathcal{S}'}(\boldsymbol{\mu}\odot\mathcal{O}_\mathcal{S}).$$

Then we show that $\Delta_\mathcal{S}\ge 0$ for any $\mathcal{S}$. Suppose that $\mathcal{S}'$ is the super arm such that

$$r_\mathcal{S}(\boldsymbol{\mu}\odot\mathcal{O}_\mathcal{S}) - r_{\mathcal{S}'}(\boldsymbol{\mu}\odot\mathcal{O}_\mathcal{S}) = \Delta_\mathcal{S}.$$

Then because any $u\notin\mathcal{S}$ will not increase the reward under the mean vector $\boldsymbol{\mu}\odot\mathcal{O}_\mathcal{S}$, we have

$$r_{\mathcal{S}'}(\boldsymbol{\mu}\odot\mathcal{O}_\mathcal{S}) = r_{\mathcal{S}\cap\mathcal{S}'}(\boldsymbol{\mu}\odot\mathcal{O}_\mathcal{S}).$$

Then, because the reward function is monotone in terms of the set $\mathcal{L}$, we have

$$\Delta_\mathcal{S} = r_\mathcal{S}(\boldsymbol{\mu}\odot\mathcal{O}_\mathcal{S}) - r_{\mathcal{S}'}(\boldsymbol{\mu}\odot\mathcal{O}_\mathcal{S}) = r_\mathcal{S}(\boldsymbol{\mu}\odot\mathcal{O}_\mathcal{S}) - r_{\mathcal{S}\cap\mathcal{S}'}(\boldsymbol{\mu}\odot\mathcal{O}_\mathcal{S}) \ge 0,$$

and we conclude the proof. $\square$

The following lemma is the main lemma for the proof of Theorem 3.9. It claims that if CUCB does not choose the target super arm $\mathcal{S}$, then there must be an arm not observed for enough times, and that arm must be observed.

**Lemma B.11.** *Suppose $\cap_{t=1}^T\mathcal{N}_t^s$ hold during the execution of Algorithm 1, and $\mathcal{S}$ is a super arm such that $\Delta_\mathcal{S} > 0$. Then for time $t$, if the Greedy oracle does not choose $\mathcal{S}$, then there must be a selected node such that a base arm corresponds to that node has $\rho_i^{(t)} > \frac{\Delta_\mathcal{S}}{4m}$.*

*Proof.* Suppose that the Greedy oracle chooses super arm $\mathcal{S}'\ne\mathcal{S}$, and denote $u'$ to be the first node picked by the Greedy oracle that does not belong to $\mathcal{S}$. Besides, we denote $A$ to be the set of nodes before $u'$ is picked, and $u$ to be a node in $\mathcal{S}\setminus A$. We denote $M_A$, $M_u$ and $M_{u'}$ to be the set of base arms connected to the node set $A$, the node $u$ and $u'$ respectively. Then we use $\mathbf{UCB}_t$ to denote the upper confidence value of the arms. Then we show that there exists a base arm $i$ connected to the node in $\{u'\}\cap A$ such that $\rho_i^{(t)} > \frac{\Delta_\mathcal{S}}{4m}$.

We prove by contradiction, suppose that there does not exist a base arm $i$ connected to the node in $\{u'\}\cap A$ such that $\rho_i^{(t)} > \frac{\Delta_\mathcal{S}}{4m}$.

We first show a lower bound on $r_{A\cup\{u\}}(\mathbf{UCB}_t) - r_A(\mathbf{UCB}_t)$. First under $\cap_{t=1}^T\mathcal{N}_t^s$, we know that $\mathbf{UCB}_t\succeq\boldsymbol{\mu}\odot\mathcal{O}_\mathcal{S}$, thus because of the monotonicity (Assumption 2.1), we have $r_{A\cup\{u\}}(\mathbf{UCB}_t)\ge r_{A\cup\{u\}}(\boldsymbol{\mu})$. Besides, because $\cap_{t=1}^T\mathcal{N}_t^s$ hold, we also have $\mathbf{UCB}_t\preceq\boldsymbol{\mu}\odot\mathcal{O}_\mathcal{S}+2\boldsymbol{\rho}^{(t)}$. Thus from Lemma B.9, we have

$$r_A(\mathbf{UCB}_t)\le r_A(\boldsymbol{\mu}\odot\mathcal{O}_\mathcal{S}+2\boldsymbol{\rho}^{(t)}) = r_A(\boldsymbol{\mu}\odot M_A+2\boldsymbol{\rho}^{(t)}\odot M_A)\le r_A(\boldsymbol{\mu})+2m\cdot\frac{\Delta_\mathcal{S}}{4m} = r_A(\boldsymbol{\mu})+\frac{\Delta_\mathcal{S}}{2}.$$

Thus we have

$$r_{A\cup\{u\}}(\mathbf{UCB}_t) - r_A(\mathbf{UCB}_t) \ge r_{A\cup\{u\}}(\boldsymbol{\mu}) - r_A(\boldsymbol{\mu}) - \frac{\Delta_\mathcal{S}}{2}$$

$$\ge \Delta_\mathcal{S} - \frac{\Delta_\mathcal{S}}{2}$$

$$= \frac{\Delta_\mathcal{S}}{2},$$

where we apply Lemma B.10 in the second inequality.

Then we show an upper bound on $r_{A\cup\{u'\}}(\mathbf{UCB}_t) - r_A(\mathbf{UCB}_t)$. Since $u'$ does not belong to the super arm $\mathcal{S}$, we have

$$r_{A\cup\{u'\}}(\mathbf{UCB}_t) - r_A(\mathbf{UCB}_t) \le r_{\{u'\}}(\mathbf{UCB}_t) \le m\frac{\Delta_\mathcal{S}}{4m} = \frac{\Delta_\mathcal{S}}{4}.$$

Thus because we assume $\Delta_\mathcal{S} > 0$, we have

$$r_{A\cup\{u\}}(\mathbf{UCB}_t) - r_A(\mathbf{UCB}_t) > r_{A\cup\{u'\}}(\mathbf{UCB}_t) - r_A(\mathbf{UCB}_t),$$

which means that the Greedy oracle will not select $u'$, and we conclude the proof. $\square$

Now with Lemma B.11, we can prove Theorem 3.9.

*Proof of Theorem 3.9.* The proof is a straightforward application of Lemma B.11. First, under $\cap_{t=1}^{T} \mathcal{N}_t^s$, which happen with probability $1 - \delta$, if CUCB does not choose $\mathcal{S}$ at round $t$, it means that there exists a base arm $i$ belongs to $\mathcal{O}_{\mathcal{S}}$ such that $\rho_i^{(t)} > \frac{\Delta_{\mathcal{S}}}{4m}$, which means that

$$\sqrt{\frac{\ln(4mT^3/\delta)}{2T_i^{(t)}}} \geq \sqrt{\frac{\ln(4mt^3/\delta)}{2T_i^{(t)}}} > \frac{\Delta_{\mathcal{S}}}{4m}.$$

Thus we have

$$T_i^{(t)} \leq \frac{8m^2 \ln(4mT^3/\delta)}{\Delta_{\mathcal{S}}^2}.$$

However after not choose $\mathcal{S}$, $T_i^{(t)}$ will increase by 1. Thus, the total number of times not choosing $\mathcal{S}$ is bounded by $\frac{8m^3 \ln(4mT^3/\delta)}{\Delta_{\mathcal{S}}^2}$ since there are at most $m$ base arms. The attack cost is bounded by $\frac{8m^4 \ln(4mT^3/\delta)}{\Delta_{\mathcal{S}}^2}$ since the attack cost at each time is bounded by $m$. Thus, there exist $T^* = \text{poly}(m, 1/\Delta_{\mathcal{S}}, \log 1/\delta)$ such that for all $T \geq T^*$,

$$\frac{8m^4 \ln(4mT^3/\delta)}{\Delta_{\mathcal{S}}^2} \leq \sqrt{T}.$$

Thus, the probabilistic maximum coverage problem with CUCB and Greedy oracle is polynomially attackable. □

## B.3 PROOF OF COROLLARY 3.10

In this part, we prove Corollary 3.10. We first state our settings, and then show how to reduce the simple RL setting to CMAB.

**Episodic MDP** We consider the episodic Markov Decision Process (MDP), denoted by the tuple $(\texttt{State}, \texttt{Action}, H, \mathcal{P}, \boldsymbol{\mu})$, where $\texttt{State}$ is the set of states, $\texttt{Action}$ is the set of actions, $H$ is the number of steps in each episode, $\mathcal{P}$ is the transition metric such that $\mathbb{P}(\cdot|s, a)$ gives the transition distribution over the next state if action $a$ is taken in the current state $s$, and $\boldsymbol{\mu} : \texttt{State} \times \texttt{Action} \to \mathbb{R}$ is the expected reward of state action pair $(s, a)$. We assume that the states $\texttt{State}$ and the actions $\texttt{Action}$ are finite. We work with the stationary MDPs here with the same reward and transition functions at each $h \leq H$, and all our analyses extend trivially to non-stationary MDPs. We assume that the transition probability $\mathcal{P}$ is known in advance.[3]

An RL agent (or learner) interacts with the MDP for $T$ episodes, and each episode consists of $H$ steps. In each episode of the MDP, we assume that the initial state $s(1)$ is fixed for simplicity. In episode $t$ and step $h$, the learner observes the current state $s_t(h) \in \texttt{State}$, selects an action $a_t(h) \in \texttt{Action}$, and incurs a noisy reward $r_{t,h}(s_t(h), a_t(h))$. Also, we have $\mathbb{E}[r_{t,h}(s_t(h), a_t(h))] = \mu(s_t(h), a_t(h))$. Our results can also be extended to the setting where the reward is dependent on step $h \leq H$.

A (deterministic) policy $\pi$ of an agent is a collection of $H$ functions $\{\pi_h : \texttt{State} \to \texttt{Action}\}$. The value function $V_h^\pi(s)$ is the expected reward under policy $\pi$, starting from state $s$ at step $h$, until the end of the episode, namely

$$V_h^\pi(s) = \mathbb{E}\left[\sum_{h'=h}^{H} \mu(s_{h'}, \pi_{h'}(s_{h'})) | s_h = s\right].$$

where $s_{h'}$ denotes the state at step $h'$. Likewise, the Q-value function $Q_h^\pi(s, a)$ is the expected reward under policy $\pi$, starting from state $s$ and action $a$, until the end of the episode, namely

$$Q_h^\pi(s, a) = \mathbb{E}\left[\sum_{h'=h+1}^{H} \mu(s_{h'}, \pi_{h'}(s_{h'})) | s_h = s, a_h = a\right] + \mu(s, a).$$

---

[3]Unlike most RL literature with unknown transition probability $\mathcal{P}$, we need to know the transition probability of the MDP, i.e., the white-box setting, for a direct reduction from this simple RL setting to CMAB and solved by CUCB. We believe similar techniques can also be applied to study the attackability of RL instances with unknown transition probability.

Since `State`, `Action` and $H$ are finite, there exists an optimal policy $\pi^*$ such that $V_h^{\pi^*}(s) = \sup_\pi V_h^\pi(s)$ for any state $s$.

The regret $\mathcal{R}^\mathcal{A}(T, H)$ of any algorithm $\mathcal{A}$ is the difference between the total expected true reward from the best fixed policy $\pi^*$ in hindsight, and the expected true reward over T episodes, namely

$$\mathcal{R}^\mathcal{A}(T, H) = \sum_{t=1}^{T} \left( V_1^{\pi^*}(s(1)) - V_1^{\pi_t}(s(1)) \right).$$

The objective of the learner is to minimize the regret $\mathcal{R}^\mathcal{A}(T, H)$.

**Reward poisoning attack (reward manipulation)**   For the reward poisoning attack, or reward manipulation, the attacker can intercept the reward $r_{t,h}(s_t(h), a_t(h))$ at every episode $t$ and step $h$, and decide whether to modify the reward $r_{t,h}(s_t(h), a_t(h))$ to $\tilde{r}_{t,h}(s_t(h), a_t(h))$. The goal of the attack process is to fool the algorithm to play a target policy $\pi$ for $T - o(T)$ times. The cost of the whole attack process is defined as $C(T) = \sum_{t=1}^{T} \sum_{h=1}^{H} \mathbb{I}\{r_{t,h}(s_t(h), a_t(h)) \neq \tilde{r}_{t,h}(s_t(h), a_t(h))\}$.

**Reduction to CMAB problem**   Note that in this simple RL setting where the transition probability is known in advance, it can be reduced to a CMAB problem and thus solved by the CUCB algorithm. We now construct the CMAB instance.

1. There are $|\text{State}| \times |\text{Action}|$ base arms, each base arm $(s, a)$ denote the random reward $r(s, a)$, with unknown expected value $\mathbb{E}[r(s, a)] = \mu(s, a)$.

2. Each policy $\pi$ is a super arm, and the expected reward of super arm $\pi$ is just the value function $V_1^\pi(s(1))$.

3. Note that when we select a policy $\pi$ (a super arm) and execute $\pi$ for this episode $t$, a random set of base arms $\tau_t$ will be triggered and observed following distribution $\mathcal{D}$, and the distribution $\mathcal{D}$ is determined by the episodic Markov Decision Process. Define $A_h^\pi(s, a)$ as the probability that it will go to state $s$ with $\pi(s) = a$ at time $h$, then the base arm $(s, a)$ is triggered with probability $\sum_{h=1}^{H} A_h^\pi(s, a)$ in each episode, thus $p_{(s,a)}^{\mathcal{D},\pi} = \sum_{h=1}^{H} A_h^\pi(s, a)$, where $p_{(s,a)}^{\mathcal{D},\pi}$ is the probability to trigger arm $(s, a)$ under policy $\pi$ (defined in Section 2).

Now the episodic MDP problem is reduced to a CMAB instance, and the remaining problem is to validate Assumption 2.1 and Assumption 2.2. Note that the reward of the super arm (policy) under the expected reward $\boldsymbol{\mu}$ is given as

$$
\begin{aligned}
r_\pi(\boldsymbol{\mu}) =& V_1^\pi(s(1)) \\
=& \mathbb{E}\left[ \sum_{h'=1}^{H} \mu(s_{h'}, \pi_{h'}(s_{h'})) | s_1 = s(1) \right] \\
=& \sum_{h'=1}^{H} \mathbb{E}\left[ \mu(s_{h'}, \pi_{h'}(s_{h'})) | s_1 = s(1) \right] \\
=& \sum_{h'=1}^{H} \sum_{s,a} \mu(s, a) A_h^\pi(s, a) \\
=& \sum_{s,a} \mu(s, a) \sum_{h'=1}^{H} A_h^\pi(s, a) \\
=& \sum_{s,a} \mu(s, a) p_{(s,a)}^{\mathcal{D},\pi}.
\end{aligned}
$$

Thus, Assumption 2.1 and Assumption 2.2 hold naturally, and we can apply Theorem 3.6 to determine if an episodic MDP is polynomially attackable or not.

## C    MISSING PROOFS IN SECTION 4

In this section, we prove Theorem 4.1. We first restate the theorem as below.

**Theorem 4.1.** *There exists a CMAB instance satisfying Assumption 2.1 and 2.2 such that it is polynomially attackable given the parameter $\boldsymbol{\mu}$ (induced from the instance's base arms' joint distribution $\mathcal{D}$), but there exists no attack algorithm that can efficiently attack the instances for* CUCB *algorithm with unknown parameter $\boldsymbol{\mu}$.*

As presented in Section 4, the hardness result for CMAB instances comes from the combinatorial structure of the super arms, which may "block" the exploration for other base arms. However when the attacker does not know the vector $\boldsymbol{\mu}$, she needs to observe different base arms to get some estimation of the problem parameter $\boldsymbol{\mu}$, and choose a super arm $\mathcal{S}$ to attack. The following section formalizes this idea.

Before we go to the proof, we restate the hard instances we constructed (Example 4.2) as follows.

**Example C.1** (Hard example). *We construct the following CMAB instance $\mathcal{I}_i$. There are $2n$ base arms, $\{a_i\}_{i\in[n]}$ and $\{b_i\}_{i\in[n]}$, and each corresponds to a random variable ranged in $[0,1]$. We have $\mu_{a_j} = 1 - 2\epsilon$ for all $j \neq i$ and $\mu_{a_i} = 1$, and $\mu_{b_j} = 1 - \epsilon$ for all $j \in [n]$ for some $\epsilon > 0$. Then we construct $n + 2$ super arms. $\mathcal{S}_j$ for all $j \in [n]$ will observe base arms $a_j$ and $b_j$, and $r_{\mathcal{S}_j} = \mu_{a_j} + \mu_{b_j}$. There is another super arm $\mathcal{S}_{n+1}$ that will observe base arms $b_j$ for all $j \in [n]$, and $r_{\mathcal{S}_{n+1}} = \sum_{j\in[n]} \mu_{b_j} + (1 - \epsilon)$. Besides, there is also a super arm $\mathcal{S}_0$ with constant reward $\mathcal{S}_0 = 2 - 2\epsilon$ and does not observe any base arm. Then, the attack super arm set $\mathcal{M} = \{\mathcal{S}_j\}_{j\in[n]}$. In total, we can construct $n$ hard instances $\mathcal{I}_i$ for all $i \in [n]$.*

The following lemma claim that the instances we construct are actually polynomially attackable (Definition 3.1 by computing the Gap (Definition 3.5).

**Lemma C.2.** *For instance $\mathcal{I}_i$, we have $\Delta_{\mathcal{S}_i} = \epsilon > 0$ and $\Delta_{\mathcal{S}_j} = -\epsilon < 0$ for all $j \neq i, j \in [n]$.*

*Proof.* First recall the definition of Gap (Definition 3.5) of a super arm $\mathcal{S} \in \mathcal{M}$ is defined as

$$\Delta_{\mathcal{S}} := r_{\mathcal{S}}(\boldsymbol{\mu}) - \max_{\mathcal{S}' \neq \mathcal{S}} r_{\mathcal{S}'}(\boldsymbol{\mu} \odot \mathcal{O}_{\mathcal{S}}).$$

Now for the instance $\mathcal{I}_i$, we have

$$\begin{aligned}
\Delta_{\mathcal{S}_i} &= r_{\mathcal{S}_i}(\boldsymbol{\mu}) - \max_{\mathcal{S}_{i'} \neq \mathcal{S}_i} r_{\mathcal{S}_{i'}}(\boldsymbol{\mu} \odot \mathcal{O}_{\mathcal{S}_i}) \\
&= r_{\mathcal{S}_i}(\boldsymbol{\mu}) - r_{\mathcal{S}_0}(\boldsymbol{\mu}) \\
&= (\mu_{a_i} + \mu_{b_i}) - (2 - 2\epsilon) \\
&= 1 + (1 - \epsilon) - (2 - 2\epsilon) \\
&= \epsilon.
\end{aligned}$$

Similarly, we have

$$\begin{aligned}
\Delta_{\mathcal{S}_j} &= r_{\mathcal{S}_j}(\boldsymbol{\mu}) - \max_{\mathcal{S}_{j'} \neq \mathcal{S}_j} r_{\mathcal{S}_{j'}}(\boldsymbol{\mu} \odot \mathcal{O}_{\mathcal{S}_j}) \\
&= r_{\mathcal{S}_j}(\boldsymbol{\mu}) - r_{\mathcal{S}_0}(\boldsymbol{\mu}) \\
&= (\mu_{a_j} + \mu_{b_j}) - (2 - 2\epsilon) \\
&= (1 - 2\epsilon) + (1 - \epsilon) - (2 - 2\epsilon) \\
&= -\epsilon.
\end{aligned}$$

$\square$

The next two lemmas (Lemma C.3 and C.4) show that, if the attacker let the CUCB algorithm to observe $l$ different super arms in the attack set $\mathcal{M}$, then the attacker needs to suffer exponential cost $(1/\epsilon)^{\Omega(l)}$.

**Lemma C.3.** *For a specific base arm $a$, suppose that during the attack of* CUCB*, there exists two rounds $t_1 < t_2$ such that $UCB_{t_1}(a) \leq \epsilon$ at time $t_1$ and has already been observed by $x$ times, and $UCB_{t_2}(a) \geq 1 - 2\epsilon$ at time $t_2$, then $a$ should be observed for at least $x/(2\epsilon)$ times during $(t_1, t_2)$ for $\epsilon < 1/8$.*

*Proof.* Note that $\text{UCB}_{t_1}(a) \leq \epsilon$, which means that $\tilde{\mu}_{a,t_1} \leq \epsilon$ and $\rho_{a,t_1} \leq \epsilon$. Then since $\rho_{a,t}$ will not increase and $t_2 > t_1$, we have

$$\tilde{\mu}_{a,t_2} \geq \text{UCB}_{t_2}(a) - \rho_{a,t_2} \geq \text{UCB}_{t_2}(a) - \rho_{a,t_1} = 1 - 2\epsilon - \epsilon = 1 - 3\epsilon.$$

Now suppose $a$ has been observed for $x'$ times during $(t_1, t_2]$ and $x$ times during $[0, t_1]$, we have

$$\tilde{\mu}_{a,t_2} \leq \frac{\tilde{\mu}_{a,t_1} \cdot x + 1 \cdot x'}{x + x'} \leq \frac{\epsilon \cdot x + 1 \cdot x'}{x + x'}.$$

We know that $\tilde{\mu}_{a,t_2} \geq 1 - 3\epsilon$, and thus

$$\frac{\epsilon \cdot x + 1 \cdot x'}{x + x'} \geq 1 - 3\epsilon \Rightarrow x' \geq \frac{1 - 4\epsilon}{\epsilon} x \geq \frac{1}{2\epsilon} x,$$

and we conclude the proof. □

**Lemma C.4.** *For any instance $\mathcal{I}_i$, suppose that there exist $l$ different time $t_1, \ldots, t_l$ such that* CUCB *pulls $\mathcal{S}_{t_j}$ at $t_j$ and $\mathcal{S}_{t_j}$ are different for $j \in [l]$, then $t_l \geq (1/2\epsilon)^{l-1}$ and $\mathcal{S}_{n+1}$ is pulled by at least $(1/2\epsilon)^{l-1}$ times when $l > 1$.*

*Proof.* We prove this by induction. When $l = 1$, the argument holds trivially.

Now suppose that the argument holds when $l = k$, then for $l = k + 1$, we first apply the induction argument for time $t_1, \ldots, t_{l-1}$ and we have at time $t_{l-1}$, $\mathcal{S}_{n+1}$ has already been pulled for at least $(1/2\epsilon)^{l-2}$ times if $k > 1$, which means that $\mathcal{S}_{t_l}$ has already been observed by $(1/2\epsilon)^{l-2}$ times. If $k = 1$ $(l = 2)$, we also know that $\mathcal{S}_{t_l}$ needs to be observed by at least once, since $\text{UCB}_{t_1}(b_{t_l}) \leq \epsilon$.

Now assume that $t'_{l-1}$ and $t'_l$ satisfies that $t_{l-1} \leq t'_{l-1} < t'_l \leq t_l$ such that at time $t'_{l-1}$ CUCB pulls $\mathcal{S}_{t_{l-1}}$ and at time $t'_l$ CUCB pulls $\mathcal{S}_{t_l}$, and during the period $(t'_{l-1}, t'_{l-1})$ CUCB never pulls the super arm $\mathcal{S}_{t_{l-1}}$ and $\mathcal{S}_{t_l}$. Then because CUCB will pull $\mathcal{S}_{t_l}$ at time $t'_l$, we have $\text{UCB}_{t'_l}(b_{t_l}) \geq 1 - 2\epsilon$. Besides, at time $t'_{l-1}$, since CUCB pulls $\mathcal{S}_{t_{l-1}}$, we have $\text{UCB}_{t'_{l-1}}(b_{t_l}) \leq \epsilon$. Applying Lemma C.3, we know that $b_{t_l}$ is observed for at least $(1/2\epsilon)^{l-1}$ times during $(t'_{l-1}, t'_l)$. Also note that during $(t'_{l-1}, t'_l)$, super arm $\mathcal{S}_{t_l}$ is never pulled, and the only way to observe $b_{t_l}$ is to pull the super arm $\mathcal{S}_{n+1}$. Thus, we conclude the induction step. □

Now we can finally prove Theorem 4.1. In Lemma C.4, we already show that if the attacker lets CUCB to pull $l$ different arms in the attack set $\mathcal{M}$, the attacker needs to suffer for loss exponential in $l$. Then the intuition is that: without knowing the parameter $\boldsymbol{\mu}$, the attacker needs to visit at least $\Omega(n)$ different arms in $\mathcal{M}$ to guarantee a high probability of attack success, and thus suffer for exponential cost.

*Proof of Theorem 4.1.* We prove this by contradiction. Suppose that an attack algorithm $\mathcal{A}$ with constant $\gamma'$ successfully attacks CUCB on the random instance $\mathcal{I}$, which is uniformly chosen from $\{\mathcal{I}_1, \ldots, \mathcal{I}_n\}$ with high probability. Here we assume $\epsilon < 1/4$. Then, for $T \geq P(n, K)$ where $P(n, K)$ denotes a polynomial with variable $n, K$, we know that $\mathcal{A}$ uses at most $T^{1-\gamma'}$ budget and play the super arms in $\mathcal{M} = \{\mathcal{S}_1, \ldots, \mathcal{S}_n\}$ with at least $T^{1-\gamma'}$ times.

Now we choose $n$ such that $1/(2\epsilon)^{n/2} \geq P(n, K)$, which is possible since $P(n, K)$ can be bound by some polynomials of $n$, and we consider $T = 1/(2\epsilon)^{n/2}$. Then by Lemma C.4, we know that at time $T$, CUCB visits at most $n/2$ different arms in $\mathcal{M}$. Recall by Lemma C.2, we know that for instance $\mathcal{I}_i$, only super arm $\mathcal{S}_i$ has gap $\Delta_{\mathcal{S}_i} = \epsilon > 0$ and all other super arms $\mathcal{S}_j$ for $j \neq i, j \in [n]$ has gap $\Delta_{\mathcal{S}_j} = -\epsilon < 0$. Because $\mathcal{I}$ is uniformly chosen from $\{\mathcal{I}_1, \ldots, \mathcal{I}_n\}$, then with probability at least $1/2$, CUCB does not visit the super arm with gap $\epsilon$ at time $T$.

However if the super arm $\mathcal{S}_j$ has gap $\mathcal{S}_j = -\epsilon < 0$, from Lemma B.8, $\mathcal{S}_j$ can only be observed by $P'(n, 1/\epsilon, \log(1/\delta) \cdot T^{1-\gamma'}$ times for some polynomial $P'$ on $n, 1/\epsilon, \log(1/\delta)$, under event $\cap(\neg \mathcal{N}_t^s)$, which happens with probability $1 - \delta$. Thus, with probability at least $1/2 - \delta$, CUCB can only play super arms in $\mathcal{M}$ for $n/2 \cdot P'(n, 1/\epsilon, \log(1/\delta) \cdot T^{1-\gamma'}$ times. However, by Definition 3.1, CUCB needs to pull arms in $\mathcal{M}$ for at least $T - T^{1-\gamma'}$ times, choosing $n$ large enough will get

$$T - T^{1-\gamma'} > n/2 \cdot P'(n, 1/\epsilon, \log(1/\delta) \cdot T^{1-\gamma'},$$

since $T = 1/(2\epsilon)^{n/2}$ exponentially depends on $n$, which means that with probability at least $1/2 - \delta$, CUCB will not pull arms in $\mathcal{M}$ with at least $T - T^{1-\gamma'}$ times. Note that the number of base arms $m = 2n$, we conclude the proof. □

## D  LIMITATION

One limitation of our findings is that our attackability characterization is limited to one threat model: reward poisoning attacks. The characterization cannot be directly generalized to environment poisoning attacks Rakhsha et al. (2020); Sun et al. (2021); Xu et al. (2021); Rangi et al. (2022) where the adversary can directly change the environment such as reward function. Environment poisoning is more powerful than reward poisoning and perturbation in environment will invalidate our analysis regarding $\Delta_{\mathcal{M}}$. On the other hand, our attackability characterization can be applied to action poisoning attacks Liu & Lai (2020), which in principle is still reward poisoning where the reward cannot be arbitrarily changed but has to be replaced by another action's reward. We leave the study of attackability of CMAB under other threat models as future work.

