# OpenReview forum: "Adversarial Attacks on Combinatorial Multi-Armed Bandits"
_ICLR.cc/2024/Conference — Submitted to ICLR 2024_

### Official Review · Reviewer_cTNV · 2023-10-28

**Soundness:** 2 fair
**Presentation:** 3 good
**Contribution:** 2 fair
**Rating:** 3
**Confidence:** 4

**Summary:**

This paper considers reward poisoning attacks on Combinatorial Multi-armed Bandits (CMAB), and provides a sufficient and necessary condition for the attackability of CMAB. This condition depends on the intrinsic properties of the corresponding CMAB instance such as the reward distributions of super arms and outcome distributions of base arms. The paper further illustrates that the attackability of a specific CMAB instance also depends on whether the bandit instance is known or unknown to the adversary.

**Strengths:**

The paper considers the adversarial attacks on combinatorial multi-Armed bandit. The paper introduces a new notion of attackability, which has a stronger requirement than existing conditions. The paper also characterizes a necessary and sufficient condition for this attackability when the underlying CMAB instance in a known environment.

**Weaknesses:**

1. The paper focuses on the new notion of attackability that requires the cost to scale polynomial in $m$. This is not well motivated. The paper only mentioned that "in practice, the exponential cost in $m$ can exceed $T$, resulting in vacuous results." Note that $T$ is growing, and we care about how the regret and the attack costs grow in terms of $T$. On the other hand, $m$ is fixed. So why it is more important to focus on the scaling in terms of $m$ than the scaling in terms of $T$?

2. The paper focuses mostly on the polynomial attackability of a CMAB instance in a known environment, i.e., all parameters of the instance such as the reward distributions of super arms and outcome distributions of base arms are given. This white box setting is of limited interest for practice.

**Questions:**

1. Better justify why one should focus on the scaling of $m$ term.
2. Can the authors provide the corresponding conditions for the black-box setting?

---

> ### Author Response · Authors · 2023-11-21
> **Response to Reviewer cTNV**
>
> Thank you for the reviews and questions. We provide a detailed response to your comments as follows.
>
> **Q.** The paper focuses on the new notion of attackability that requires the cost to scale polynomial in $m$. This is not well motivated. The paper only mentioned that "in practice, the exponential cost in $m$ can exceed $T$, resulting in vacuous results." Note that $T$ is growing, and we care about how the regret and the attack costs grow in terms of $T$. On the other hand, $m$ is fixed. So why it is more important to focus on the scaling in terms of $m$ than the scaling in terms of $T$?
>
>
> **A.** We thank the reviewer for raising this issue. We think the question has two parts: 1. why consider the dependency on $m$, and 2. why not try to improve the dependency on $T$.
>
> *As for the part 1*, we believe it is not only for the attack, but for the whole bandit theory. Note that the regret of the most basic multi-armed bandit also scales with the number of arms $K$, say $\tilde O(\sqrt{KT})$ for distribution-independent regret bound, and $\tilde O(K\log T)$ for distribution-dependent regret bound. However, people tried hard to improve the dependency on with respect to $K$, since in reality, people cannot have an infinite time horizon $T$. Thus, people considered many structures for the bandit instances, like linear (contextual) bandit [1] and combinatorial semi-bandit [2,3,4], which can have exponential number of arms. Without leveraging the internal structures, the regret bound will be $\tilde O(\exp(K) \sqrt{T})$ for distribution-independent bound and $\tilde O(\exp(K)\log T)$ for distribution-dependent bound, and when $T$ is not that large (i.e., exponential in $K$), the bounds are vacuous. These results highlight the importance of considering the dependency besides $T$.
>
> Similar to the concern in the regret analysis in bandit, when considering the attack cost, people do not have infinite cost for the attack, and thus, constraining the attack cost to be polynomial dependent on different terms is quite important.
>
> *As for the part 2*, it is a really good question. Note that we aim to get a unified theory to attack any potential bandit algorithm with sublinear regret, and thus the exact attack cost with respect to $T$ will be dependent on the victim algorithm. As shown in Corollary 3.7, if the victim algorithm is CUCB, the attack cost is already $O(\log T)$. Although it might be possible to even reduce the dependency, we believe that $O(\log T)$ is already satisfactory and thus it is more important to reduce the dependency on $m$, i.e., reducing it from exponential dependency to polynomial dependency.
>
>
> [1] Lihong Li, Wei Chu, John Langford, and Robert E. Schapire. A contextual-bandit approach to personalized news article recommendation. In Proceedings of the 19th international conference on World wide web (pp. 661-670).
>
> [2] Wei Chen, Yajun Wang, and Yang Yuan. Combinatorial multi-armed bandit: General framework and applications. ICML 2013.
>
> [3] Wei Chen, Yajun Wang, Yang Yuan, and Qinshi Wang. Combinatorial multi-armed bandit and its extension to probabilistically triggered arms. The Journal of Machine Learning Research, 17(1): 1746–1778, 2016
>
> [4] Branislav Kveton, Zheng Wen, Azin Ashkan, Hoda Eydgahi, and Brian Eriksson. Matroid bandits: Fast combinatorial optimization with learning. arXiv preprint arXiv:1403.5045, 2014.
>
>
> **Q.** The paper focuses mostly on the polynomial attackability of a CMAB instance in a known environment, i.e., all parameters of the instance such as the reward distributions of super arms and outcome distributions of base arms are given. This white box setting is of limited interest for practice.
> Can the authors provide the corresponding conditions for the black-box setting?
>
> **A.** We thank the reviewer for raising the concern. While it is true that  the black-box setting is more interesting, it is also harder to study. As the first study on the attackability of CMAB, we focus on analyzing the white-box setting and show that it is difficult to attack such an environment even when all parameters are known. We believe the characterization is highly non-trivial and significantly different from white-box setting considering the hardness example, where we showed an instance can be polynomially attackable when the the environment is known but becomes polynomially unattackable when the environment is unknown.
>
> On the other hand, our attack algorithm is already applied to several applications under black-box setting, such as probabilistic max coverage, online minimum spanning tree and cascading bandits where the problems are always attackable. We can directly apply Algorithm 1 even under black-box setting for these applications. We emphasize that our experiments in Figure 2 (a-d), (g, h) are all attacked by Algorithm 1 under black-box setting.

---

### Official Review · Reviewer_grhj · 2023-10-30

**Soundness:** 2 fair
**Presentation:** 2 fair
**Contribution:** 2 fair
**Rating:** 5
**Confidence:** 2

**Summary:**

This work studies adversarial attacks on combinatorial bandits (CMAB).

**Strengths:**

Adversarial attack has been studied in stochastic bandits, linear bandits, adversarial bandits. The study of adversarial attack on CMAB is new.
This work proposes new notions of attackability based on the structure of CMAB.

**Weaknesses:**

While the framework follows from previous work Wang & Chen, I feel the paper could benefit from a discussion on simpler CMAB models first (i.e. without the trigger function etc. )

**Questions:**

I did not quite follow the problem setup. The formulation follows from the previous work Wang & Chen, which used infinite action space with the trigger function. However, in this submission, in the definition of super arms, it is first mentioned the action space could be infinite. But it also mentions each super arm is a set of base arm, which implies the cardinality of super arms is 2^m. Also, it seems that if we define a super arm as a set of base arms, then there is no need to define the trigger function?

Given my unfamiliarity with this line of work, it is currently unclear to me exactly how the action set is defined in this current work.

---

> ### Author Response · Authors · 2023-11-21
> **Response to Reviewer grhj**
>
> Thank you for recognizing the novelty of our paper and we appreciate your reviews and questions. We provide a detailed response to your comments as follows.
>
> **Q.** I did not quite follow the problem setup. The formulation follows from the previous work Wang & Chen, which used infinite action space with the trigger function. However, in this submission, in the definition of super arms, it is first mentioned the action space could be infinite. But it also mentions each super arm is a set of base arm, which implies the cardinality of super arms is 2^m. Also, it seems that if we define a super arm as a set of base arms, then there is no need to define the trigger function?
>
> **A.** We thank the reviewer for the concerns raised. Firstly, we would like to apologize for and clarify the confusion here.
>
> We define the action such that it will trigger a set of base arms (with certain triggering probability function). Because the triggering probability might be complicated, the total number of actions might exceed $2^m$. Here we take the shortest path and cascading bandit as an example for better understanding.
>
> In shortest path, each action corresponds to a possible path in the graph. There is no triggering function in shortest path and the total number of paths is indeed bounded by $2^m$. As for the cascading bandit, each action corresponds to an ordered list of $K$ items (here $K$ is the max number of items to show). In this case, the total number of actions is computed as $A_m^K = m * (m-1) * \cdots * (m-K+1)$. This number can be larger than $2^m$.
>
> Besides, the triggering probability is needed even if the number of actions is finite. Let's still use the cascading bandit as an example. Here, the triggering function would define whether the next item in the list is observed, as the user stops examining items after choosing his or her first attractive item. Thus even if the action is exactly the same, say we list item 1,2,3 in order and let the users to choose. Sometimes we only know that item 1 is chosen while not observing any reward for item 2 and 3 (thus only observe 1 item), while sometimes we know that 1 and 2 are not chosen by some user and the user chooses item 3 (thus observe 3 items).
>
> **Q.** Given my unfamiliarity with this line of work, it is currently unclear to me exactly how the action set is defined in this current work.
>
> **A.** Thanks for your question, please refer to our answer to your first question.

---

### Official Review · Reviewer_UBzx · 2023-11-01

**Soundness:** 2 fair
**Presentation:** 2 fair
**Contribution:** 2 fair
**Rating:** 6
**Confidence:** 4

**Summary:**

This paper explores adversarial attacks on Combinatorial Multi-Armed Bandits (CMAB). It discusses a sufficient and necessary condition for the polynomial attackability of CMAB and presents an attack algorithm for attackable instances. The authors also investigate how the attackability of a CMAB instance is influenced by whether the bandit instance is known or unknown to the adversary, which indicates that adversarial attacks on CMAB are difficult in practice and a general attack strategy for any CMAB instance does not exist. The findings are validated through experiments on real-world CMAB applications.

**Strengths:**

Originality: The paper introduces a novel concept of polynomial attackability in the context of combinatorial multi-armed bandits (CMAB). This notion captures the vulnerability and robustness of CMAB systems, which is a unique contribution to the field.

Quality: The paper provides a rigorous analysis of the attackability of CMAB systems and presents a sufficient and necessary condition for polynomial attackability. The paper also presents an attack algorithm for attackable instances. The paper validate the theoretical findings and demonstrate the effectiveness of the proposed attack via extensive experiments conducted on various CMAB applications.

Clarity: The paper is well-written and presents the concepts, definitions, and analysis in a clear and concise manner. The experimental setup and results are explained in detail, and the source code is provided, making it easy for readers to understand and replicate the experiments.

Significance: The paper addresses an important research question regarding the vulnerability of CMAB systems to adversarial attacks. By introducing the concept of polynomial attackability and providing a comprehensive analysis, the paper contributes to the understanding of the security and robustness of CMAB algorithms. The finding that the attackability of a specific CMAB instance also depends on whether the bandit instance is known or unknown to the adversary is impressive and may have practical implications for designing more secure CMAB systems in real-world applications.

**Weaknesses:**

1. The limitations of the findings are less discussed.

The findings regard the polynomial attackability highly depends on the threat model, in which the outcome of the base arms can be modified by the adversary. However, recent researchers discussed different types of adversarial attacks on bandit and RL [1-5], including also environment poisoning attack and action poisoning attack. In the CMAB system, the environment-manipulation adversary could manipulate the reward function $r$ and the action-manipulation adversary could manipulate the super arm $S$.

The proposed sufficient and necessary condition of the polynomial attackability is limited to the specific reward-manipulation adversary. If one environment-manipulation adversary can manipulate the reward function, $\Delta_M$ could be changed and the condition of the polynomial attackability does not work. Example 4.2 (Hard example) is also limited to the specific reward-manipulation adversary. Some statement in the abstract and introduction is inaccurate. The limitations of the findings are less discussed.

A more thorough discussion about the scope and limitation of the findings would be helpful.

2. Some problems in experimental results.

I found that the numerical experiments do not reflect the effect of the proposed attack algorithm. For example, in (2(g), 2(h)), the number of the target arm pulls is at most 4e3 after 1e5 iterations. The target arm is pulled in only 4% rounds.

In addition, the experiment that reflects the polynomial unattackablity would be helpful. I recommend that the author can run some experiments on the hard example. The adversary can attack the hard example instance using heuristics with sublinear attack budget.

[1] Amin Rakhsha, Goran Radanovic, Rati Devidze, Xiaojin Zhu, and Adish Singla. Policy teaching via environment
poisoning: Training-time adversarial attacks against reinforcement learning. In International Conference on
Machine Learning, pages 7974–7984, 2020.

[2] Guanlin Liu and Lifeng Lai, Action-Manipulation Attacks Against Stochastic Bandits: Attacks and Defense, in IEEE Transactions on Signal Processing, vol. 68, pp. 5152-5165, 2020

[3] Yanchao Sun, Da Huo, and Furong Huang. Vulnerability-aware poisoning mechanism for online rl with unknown
dynamics. In International Conference on Learning Representations, 2021.

[4] Hang Xu, Rundong Wang, Lev Raizman, and Zinovi Rabinovich. Transferable environment poisoning: Training-time attack on reinforcement learning. In Proceedings of the 20th international conference on autonomous agents and multiagent systems, pages 1398–1406, 2021.

[5] Anshuka Rangi, Haifeng Xu, Long Tran-Thanh, and Massimo Franceschetti. Understanding the limits of
poisoning attacks in episodic reinforcement learning. In Proceedings of the Thirty-First International Joint Conference on Artificial Intelligence, IJCAI-22, pages 3394–3400, 2022.

**Questions:**

Overall, I like this paper but some statement in the abstract and introduction is inaccurate and the limitations of the findings are less discussed. Could the author provide more discussion about the scope and limitation of the findings?

---

> ### Author Response · Authors · 2023-11-21
> **Response to Reviewer UBzx**
>
> Thank you for your detailed comments. Your recognition of the novelty and significance of our work is highly appreciated. We provide a detailed response to your comments as follows.
>
> **Q.** The limitations of the findings are less discussed.
> The findings regard the polynomial attackability highly depends on the threat model, in which the outcome of the base arms can be modified by the adversary. However, recent researchers discussed different types of adversarial attacks on bandit and RL [1-5], including also environment poisoning attack and action poisoning attack. In the CMAB system, the environment-manipulation adversary could manipulate the reward function $r$ and the action-manipulation adversary could manipulate the super arm $\mathcal{S}_i$.
>
> The proposed sufficient and necessary condition of the polynomial attackability is limited to the specific reward-manipulation adversary. If one environment-manipulation adversary can manipulate the reward function, $\Delta_\mathcal{M}$ could be changed and the condition of the polynomial attackability does not work. Example 4.2 (Hard example) is also limited to the specific reward-manipulation adversary. Some statement in the abstract and introduction is inaccurate. The limitations of the findings are less discussed.
>
> A more thorough discussion about the scope and limitation of the findings would be helpful.
>
> **A.** We appreciate your suggestions on discussing other threat models such as environment poisoning attacks and action poisoning attacks. In the revised version, we have discussed other threat models in related works. We have clearly stated in abstract and introduction that the threat model in our paper is reward poisoning attacks. We would like to emphasize that reward poisoning attack is arguably one of the most popular threat models in adversarial attacks against bandits and RL. As the first paper studying attackability of combinatorial bandits, we focused on reward poisoning and left other threat models as future work. We also added discussion about the limitation of the work in Appendix D, where we offered insights on generalizing attackability condition to other threat models. Please see our revision for details.
>
>
> **Q.** Some problems in experimental results. I found that the numerical experiments do not reflect the effect of the proposed attack algorithm. For example, in (2(g), 2(h)), the number of the target arm pulls is at most 4e3 after 1e5 iterations. The target arm is pulled in only 4% rounds.
>
> **A.** Thank you for your detailed comments on experiments. We have performed additional experiments on cascading bandits and reported the results in Appendix A.2. We compared three settings with different number of items $K$ and items to rank $m$, including (1) $m = 25, K = 5000$, which is the original setting in Figure 2(g), 2(h), (2)$m = 35, K = 1000$, and (3) $m = 5, K = 1000$. We observe that with a smaller action space, the target arm is pulled around 50% rounds in 10K iterations for CascadeKLUCB with $m = 35, K = 1000$ in Figure 4(d), and the number increased to 90% rounds $m = 5, K = 1000$ in Figure 4(f). This trend suggested that with longer iterations the CMAB algorithm will pull target arm more often. The results also validated our theoretical analysis.
>
> **Q.** In addition, the experiment that reflects the polynomial unattackablity would be helpful. I recommend that the author can run some experiments on the hard example. The adversary can attack the hard example instance using heuristics with sublinear attack budget.
>
> **A.** We thank the reviewer for raising the concern, and we point the reader towards Figure 2(e) and 2(f). There we showed that in an unattackable case of online shortest path, while the adversary keeps attacking the CMAB algorithm and spends *linear* budget, the target arm is still not pulled.

---

> > ### Comment · Reviewer_UBzx · 2023-11-21
> >
> > My major concerns are addressed and thus I increased my score to reflect that. I recommend that the authors put the experimental results of Appendix A.2 into the main page. Some subfigures in Figure 2 cannot correctly illustrate the efficacy of the proposed algorithm.

---

### Official Review · Reviewer_eFM9 · 2023-11-01

**Soundness:** 3 good
**Presentation:** 2 fair
**Contribution:** 2 fair
**Rating:** 6
**Confidence:** 4

**Summary:**

This paper delves into adversarial attacks targeting combinatorial multi-armed bandits (CMAB). The authors introduce the concept of polynomial attackability in CMAB, wherein an attack is deemed successful if its cost remains sublinear with respect to the time horizon and polynomial, rather than exponential, in relation to the number of base arms. They provide both sufficient and necessary conditions for such polynomial attackability within CMAB and introduce an efficient attack algorithm. The discourse further extends to the challenges of polynomial attackability for CMAB instances in unknown environments, emphasizing the absence of a universal attack approach that guarantees success with polynomial costs under such unknown CMAB instances. Empirical results across diverse CMAB scenarios validate their theoretical findings.

**Strengths:**

1) This paper is the first to study adversarial attacks against CMAB algorithms, which is an interesting and timely topic.
2) The novel characterization of the sufficient and necessary conditions for polynomial attackability in CMAB provides insight into the distinct challenges posed by CMAB instances with polynomial costs.
3) The author presents a hard example highlighting that an instance can be polynomially attackable when the adversary is aware of the environment but becomes polynomially unattackable when the environment is unknown. This underscores the difficulty of launching general adversarial attacks on unknown CMAB instances.

**Weaknesses:**

1) Algorithm 1 seems to be straightforward but may lead to large attack costs when $\Delta$ is small. The attack cost's dependency on $\Delta$ from previous works [Jun et al., 2018, Liu and Shroff, 2019] is usually linear in $\sum_{i}  \Delta_i$, while the dependency in this paper is $1 / \Delta_{S^*}$, which is worse than $\sum_{i}  \Delta_i$ as $\Delta_i \le 1$. This is due to the lack of fine-grained attack value design.

2) While Theorem 4.1 establishes the difficulty of successfully targeting general unknown CMAB instances, there remains a potential to execute attacks in particular CMAB settings, such as PMC with **unknown** base arms. This is important since learning from the attacker side is one of the main challenges of attack design in previous works: simple oracle attacks [Jun et al., 2018, Liu and Shroff, 2019] can easily attack known K-armed bandit instances. I would expect more algorithm design and analysis for the unknown environment.

**Questions:**

1) In Corollary 4.3, there is no guarantee that the randomly picked super arm $\mathcal{S}$ satisfied $\Delta_{\mathcal{S}} > 0$ (even for cascading bandits, online MST, and online PMC problem with greedy oracle, there exists $\mathcal{S}$ such that $\Delta_{\mathcal{S}} = 0$).

---

> ### Author Response · Authors · 2023-11-21
> **Response to Reviewer eFM9 (Part 1)**
>
> Thank you for your detailed review and positive feedback. We appreciate your recognition of the importance of the study, the novelty of our characterization and the hardness example. We provide a detailed response to your comments as follows.
>
> **Q.** Algorithm 1 seems to be straightforward but may lead to large attack costs when $\Delta$ is small. The attack cost's dependency on $\Delta$ from previous works [Jun et al., 2018, Liu and Shroff, 2019] is usually linear in $\sum_i \Delta_i$, while the dependency in this paper is $1/\Delta_s$, which is worse as $\Delta_i \leq 1$. This is due to the lack of fine-grained attack value design.
>
> **A.**  Thanks for this insightful question. We would like to emphasize that our attack cost does not fully come from the lack of fine-grained attack value design, but instead, it mainly comes from different settings for consideration and different definitions of the $\Delta$.
>
> + In [Jun et al,. 2018, Liu and Shroff, 2019], any real number reward is considered feasible, and the authors assume the reward distributions are $\sigma$-subgaussian. In our setting, each base arms can only take value from $[0,1]$. This different assumption on the environment makes the attacks very different: In [Jun et al,. 2018, Liu and Shroff, 2019], it is possible to give any reward value back, while in our setting, the lowerest reward we can give is 0.
> + The definition on $\Delta$ is different in [Jun et al,. 2018, Liu and Shroff, 2019] and in our paper. Let's take the bandit environment with two arms as an example, and the expected reward of two arms are $\mu_1, \mu_2$ ($\mu_1 > \mu_2$). Here $\mu_2$ is the target arm. In [Jun et al,. 2018, Liu and Shroff, 2019], $\Delta_i$ is defined as the mean reward difference between arm $i$ and the target arm (which is the standard definition in bandit literature), and in the example, it will be $\mu_1 - \mu_2$; in our setting, $\Delta$ is defined as the gap between the reward of the target arm and the best of other arms **after changing the reward to 0**. In the example, $\mu_1$ is attacked to $0$, and the gap $\Delta = \mu_2$.
>
> We will then explain why these two differences lead to different attack costs in [Jun et al,. 2018, Liu and Shroff, 2019] and our paper. First, we want to mention that the attack cost in [Jun et al,. 2018, Liu and Shroff, 2019] is not exactly $\tilde O(\sum\Delta_i)$, instead, it is $\tilde O(\sum\Delta_i + 1)$ (Corollary 1, 2 in [Jun et al,. 2018]). This $\tilde O(1)$ is important for small $\Delta$. Now let's consider the case where our algorithm leads to large attack cost in our setting: also consider two arm bandit instance, where there are two arms, $\mu_1 = 2\epsilon, \mu_2 = \epsilon$, $\epsilon>0$ but is very small, and arm 2 is the target arm. In [Jun et al,. 2018], the attacker can assign reward $-1$ for the first arm, while in our setting, the worst possible reward we can assign to arm 1 will be $0$. In this case, you need to at least pay $\tilde O(1/\epsilon)$ attack cost to let the algorithm differentiate arm 1 and 2, since it is best to assign arm 1 with reward 0 and in this case the algorithm (UCB) will observe arm 1 (the non-target arm) for at least $1/\epsilon^2$ times with each time paying $\epsilon$ attack cost.
>
> Also, to make a fair comparison with the setting in [Jun et al,. 2018], we should look at the case where $\mu_1 = 2\epsilon+1/2, \mu_2 = \epsilon+1/2$. For this instance, our attack can also assign a relatively small reward to the non-target arm, since in our setting, $\Delta=1/2+\epsilon$. Thus, our attack strategy will also have cost $\tilde O(1)$.
>
> Thus in conclusion, we believe that the different form of attack cost does not come from the lack of attack value design. Instead, it mainly comes from the different abilities of the attacker (if the attacker can assign any reward value) and different definitions of the notation. We believe that in [Jun et al,. 2018, Liu and Shroff, 2019], the fine-grind attck value design is crucial in their setting since the hard case is to have big gap, while in our setting, the fine-grind attack value design may not be that important, since we have bounded reward assumption in each base arm and thus the reward will not be that large, and the infeasibility result (the accurate characterization of polynomially attackable/unattackable) is a more valuable contribution compared with the exact cost.

---

> > ### Author Response · Authors · 2023-11-21
> > **Response to Reviewer eFM9 (Part 2)**
> >
> > **Q.** While Theorem 4.1 establishes the difficulty of successfully targeting general unknown CMAB instances, there remains a potential to execute attacks in particular CMAB settings, such as PMC with unknown base arms. This is important since learning from the attacker side is one of the main challenges of attack design in previous works: simple oracle attacks [Jun et al., 2018, Liu and Shroff, 2019] can easily attack known K-armed bandit instances. I would expect more algorithm design and analysis for the unknown environment.
> >
> > **A.**  We agree that it is important to further investigate the attackability of CMAB problems in the unknown setting. However, we find that it is generally hard. First for cascading bandits, online MST, and online PMC problem with Greedy oracle, because by the definitions of the problems, every possible super arm $S$ has non-negative gaps $\Delta_S \geq 0$, which makes it almost always attackable in practice. Besides, we also found that it is also possible to reduce our hard example (Example 4.2) to a generalized version of online shortest path problem: for each edge $e$, the cost may come from a general interval $[a_e, b_e]$ instead of $[0, 1]$. We currently do not know any CMAB problem (including the original online shortest path problem) that lies in the middle of problems with $\Delta \geq 0$ (cascading bandit, online MST) and unattackable problems in unknown setting.
> >
> > **Q.** In Corollary 4.3, there is no guarantee that the randomly picked super arm $\mathcal{S}$ satisfied $\Delta_\mathcal{S}>0$(even for cascading bandits, online MST, and online PMC problem with greedy oracle, there exists $\mathcal{S}$ such that $\Delta_\mathcal{S}=0$).
> >
> > **A.** $\Delta_\mathcal{S}=0$ is a very special case and we do not consider it in the corollary. In order to analyze the $\Delta_\mathcal{S}=0$ case, we need to understand how the algorithm *breaks the tie* when two superarms have the same reward and this information is generally unknown.

---

### Meta-Review · Area_Chair_e8Zf · 2023-12-06

**Metareview:**

This paper studies adversarial attacks on combinatorial bandits. It is well written, for readers that are knowledgeable about this topic. The attacks are analyzed and evaluated empirically. The main issue with this paper are many assumptions to derive the results:

* **Monotonicity** and **smoothness** are common in combinatorial bandits and not a big deal.

* **Known model of the environment** is a strong assumption. Even the learning agent does not know it. On the other hand, if the model was known by somebody, why would someone else try to learn it online? The authors elaborated on this issue in their rebuttal. In particular, they argue that their algorithm can be applied to the black-box setting and show good experimental results. This indicates that the black-box setting analysis, which would be much more realistic, may be possible.

This paper is well executed but the white-box setting in the analysis is highly unrealistic. With this in mind, this paper is a borderline and can go either way.

**Justification For Why Not Higher Score:**

The assumption of a known environment in the bandit setting is unrealistic and discounts this work significantly.

**Justification For Why Not Lower Score:**

N/A

---

### Decision · Program_Chairs · 2024-01-16

Reject